# A DNN Optimizer that Improves over AdaBelief by Suppression of the Adaptive Stepsize Range

**Guoqiang Zhang**                                                              *g.z.zhang@exeter.ac.uk*
*Department of Computer Science*
*University of Exeter, UK*

**Kenta Niwa**                                                                    *kenta.niwa@ntt.com*
*NTT Communication Science Laboratories, Japan*

**W. Bastiaan Kleijn**                                                       *bastiaan.kleijn@vuw.ac.nz*
*School of Engineering and Computer Science*
*Victoria University of Wellington, New Zealand*

**Reviewed on OpenReview:** *https://openreview.net/forum?id=VI2JjIfU37&noteId=8STsR7LV7Y*

## Abstract

We make contributions towards improving adaptive-optimizer performance. Our improvements are based on suppression of the range of adaptive stepsizes in the AdaBelief optimizer. Firstly, we show that the particular placement of the parameter $\epsilon$ within the update expressions of AdaBelief reduces the range of the adaptive stepsizes, making AdaBelief closer to SGD with momentum. Secondly, we extend AdaBelief by further suppressing the range of the adaptive stepsizes. To achieve the above goal, we perform mutual layerwise vector projections between the gradient $\boldsymbol{g}_t$ and its first momentum $\boldsymbol{m}_t$ before using them to estimate the second momentum. The new optimization method is referred to as *Aida*. Thirdly, extensive experimental results show that Aida outperforms nine optimizers when training transformers and LSTMs for NLP, and VGG and ResNet for image classification over CIAF10 and CIFAR100 while matching the best performance of the nine methods when training WGAN-GP models for image generation tasks. Furthermore, Aida produces higher validation accuracies than AdaBelief for training ResNet18 over ImageNet. Our implementation is available at `https://github.com/guoqiang-zhang-x/Aida-Optimizer`.

## 1 Introduction

In the last decade, stochastic gradient descent (SGD) and its variants have been widely applied in deep learning LeCun et al. (2015); Vaswani et al. (2017); et al. (2016); Chen et al. (2020) due to their simplicity and effectiveness. In the literature, SGD with momentum Sutskever et al. (2013); Polyak (1964)) dominates over other optimizers for image classification tasks He et al. (2015); Wilson et al. (2017). Suppose the objective function $f(\boldsymbol{\theta}) : \boldsymbol{\theta} \in \mathbb{R}^d$ of a DNN model is differentiable. Its update expression for minimising $f(\boldsymbol{\theta})$ can be represented as

$$[\textbf{SGD with momentum}] \begin{cases} \boldsymbol{m}_t = \beta_t \boldsymbol{m}_{t-1} + \boldsymbol{g}_t \\ \boldsymbol{\theta}_t = \boldsymbol{\theta}_{t-1} - \eta_t \boldsymbol{m}_t \end{cases}, \tag{1}$$

where $\boldsymbol{g}_t = \nabla f(\boldsymbol{\theta}_{t-1})$ is the gradient at $\boldsymbol{\theta}_t$, and $\eta_t$ is the common stepsize for all the coordinates of $\boldsymbol{\theta}$. In practice, the above method is often combined with a certain step-size scheduling method for $\eta_t$ when training DNNs.

To bring flexibility to SGD with momentum, an active research trend is to introduce elementwise adaptive stepsizes for all the coordinates of $\boldsymbol{m}_t$ in (1), referred to as *adaptive optimization* Duchi et al. (2011); Tieleman & Hinton (2012); Kingma & Ba (2017). In the literature, Adam Kingma & Ba (2017) is probably the most popular adaptive optimization method (e.g., Vaswani et al. (2017); Liu et al. (2021); Zhang et al. (2019); J. Devlin & Toutanova (2018)). Its update expression can be written as

$$[\textbf{Adam}] \begin{cases} \boldsymbol{m}_t = \beta_1 \boldsymbol{m}_{t-1} + (1-\beta_1)\boldsymbol{g}_t \\ \boldsymbol{v}_t = \beta_2 \boldsymbol{v}_{t-1} + (1-\beta_2)\boldsymbol{g}_t^2 \\ \boldsymbol{\theta}_t = \boldsymbol{\theta}_{t-1} - \eta_t \frac{1}{1-\beta_1^t} \frac{\boldsymbol{m}_t}{\sqrt{\boldsymbol{v}_t/(1-\beta_2^t)}+\epsilon} \end{cases}, \tag{2}$$

where $\boldsymbol{g}_t = \nabla f(\boldsymbol{\theta}_{t-1})$, $0 < \beta_1, \beta_2 < 1$, and $\epsilon > 0$. The two vector operations $(\cdot)^2$ and $\cdot/\cdot$ are performed in an elementwise manner. The two exponential moving averages (EMAs) $\boldsymbol{m}_t$ and $\boldsymbol{v}_t$ are alternatively referred to as the first and second momentum. The two quantities $1-\beta_1^t$ and $1-\beta_2^t$ are introduced to compensate for the estimation bias in $\boldsymbol{m}_t$ and $\boldsymbol{v}_t$, respectively. $\eta_t$ is the common stepsize while $1/(\sqrt{\boldsymbol{v}_t/(1-\beta_2^t)}+\epsilon) \in \mathbb{R}^d$ represents the elementwise adaptive stepsizes.

Due to the great success of Adam in training DNNs, various extensions of Adam have been proposed, including AdamW Loshchilov & Hutter (2019), NAdam Dozat (2016), Yogi Zaheer et al. (2018), MSVAG Balles & Hennig (2017), Fromage Bernstein et al. (2020), and AdaBelief Zhuang et al. (2020). It is worth noting that in Liu et al. (2019), the authors found that better generalization could be achieved by reducing the variance of the adaptive stepsizes of Adam. In doing so, they suggested multiplying a rectified scalar by $\boldsymbol{m}_t$ when computing $\boldsymbol{\theta}_t$ in (2) when the variance is large, which is referred to as RAdam. The AdaBound method of Luo et al. (2019) is designed to avoid extremely large and small adaptive stepsizes of Adam, which has a similar effect as RAdam. In practice, AdaBound works as an adaptive method at the beginning of the training process and gradually transforms to SGD with momentum, where all the adaptive stepsizes tend to converge to a single value. Conceptually speaking, both RAdam and AdaBound aim to reduce the range of the adaptive stepsizes of Adam to mimic the convergence behavior of SGD with momentum to a certain extent.

Inspired by the above work Liu et al. (2019); Luo et al. (2019), we consider suppressing the range of adaptive stepsizes of AdaBelief. It is noted that AdaBelief extends Adam by tracking the EMA of the squared prediction error $(\boldsymbol{m}_t - \boldsymbol{g}_t)^2$. The update expressions of AdaBelief are given by Zhuang et al. (2020)

$$[\textbf{AdaBelief}] \begin{cases} \boldsymbol{m}_t = \beta_1 \boldsymbol{m}_{t-1} + (1-\beta_1)\boldsymbol{g}_t \\ \boldsymbol{s}_t = \beta_2 \boldsymbol{s}_{t-1} + (1-\beta_2)(\boldsymbol{m}_t - \boldsymbol{g}_t)^2 + \epsilon \\ \boldsymbol{\theta}_t = \boldsymbol{\theta}_{t-1} - \eta_t \frac{1}{1-\beta_1^t} \frac{\boldsymbol{m}_t}{\sqrt{\boldsymbol{s}_t/(1-\beta_2^t)}+\epsilon} \end{cases}. \tag{3}$$

We emphasise that the parameter $\epsilon$ is involved in the computation of both $\boldsymbol{s}_t$ and $\boldsymbol{\theta}_t$ in AdaBelief, which is different from that of Adam. The work of Zhuang et al. (2020) does not study the impact of $\epsilon$ in the computation of $\boldsymbol{s}_t$, and mainly focuses on the motivation of the EMA of $(\boldsymbol{m}_t - \boldsymbol{g}_t)^2$ instead of the EMA of $\boldsymbol{g}_t^2$ being employed in Adam.

In this paper, we make three contributions. Firstly, we explain why it is important to include the parameter $\epsilon$ in the computation of $\boldsymbol{s}_t$ in (3), which will be inherited by our new algorithm Aida as described later on. We show via a Taylor expansion that the inclusion of $\epsilon$ in the computation of $\boldsymbol{s}_t$ essentially suppresses the range of the adaptive stepsizes of AdaBelief. The above property makes AdaBelief closer to SGD with momentum.

Secondly, we perform layerwise vector projections to further suppress the range of adaptive stepsizes of AdaBelief. Let us denote the subvectors of $(\boldsymbol{m}_t, \boldsymbol{g}_t)$ for the $l$th layer of a DNN model as $(\boldsymbol{m}_{l,t}, \boldsymbol{g}_{l,t})$. We perform $K$ mutual-vector projections to obtain $(\boldsymbol{m}_{l,t}^{(K)}, \boldsymbol{g}_{l,t}^{(K)})$ for the $l$th layer starting from $(\boldsymbol{m}_{l,t}^{(0)}, \boldsymbol{g}_{l,t}^{(0)}) = (\boldsymbol{m}_{l,t}, \boldsymbol{g}_{l,t})$. As an extension of AdaBelief, we then track and employ the EMA (or equivalently the second momentum) of $(\boldsymbol{m}_{l,t}^{(K)} - \boldsymbol{g}_{l,t}^{(K)})^2$ for the $l$th layer, where the resulting method is referred to as *Aida*.[3] The new method has the property that the adaptive stepsizes within each neural layer have smaller statistical variance, and the layerwise average of the adaptive stepsizes are more compact across all the neural layers than the reference method. A discussion on the benefit of the above property will be provided in Subsection 3.1. In addition, a convergence analysis for Aida is conducted in Subsection 3.3. As an example, Fig. 1 and 2 demonstrate that Aida indeed produces a more compact range of adaptive stepsizes than AdaBelief and Adam for training VGG11 over CIFAR10. Furthermore, the adaptive stepsizes of Aida become increasingly more compact as the iteration increases. It is worth noting that at the end of the training process, the 11 layerwise average stepsizes in Fig. 1 do not converge to a single value, indicating the adaptability of Aida.

---

[0]As an example, considering Adam at iteration $t$, the layerwise average of adaptive stepsizes for the $l$th layer of VGG11 is computed as $\frac{1}{d_l} \sum_{i=1}^{d_l} 1/(\sqrt{\boldsymbol{v}_{l,t}[i]/(1-\beta_2^t)}+\epsilon)$, where $\boldsymbol{v}_{l,t} \in \mathbb{R}^{d_l}$ is the subvector of the 2nd momentum $\boldsymbol{v}_t \in \mathbb{R}^d$ for the $l$th layer.

[3]It is named after an Italian opera by Verdi.

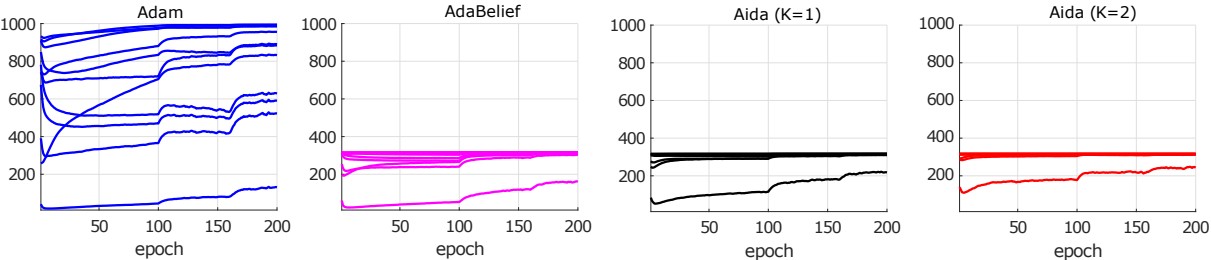

Figure 1: Comparison of layerwise average[2] of adaptive stepsizes for the 11 neural layers of VGG11 by training over CIFAR10 for 200 epochs. See Appendix C for the parameter setups of the three methods, where the optimal parameter $\epsilon$ for Adam was selected from a discrete set to give the best validation performance. The jumps in the curves at 100 and 160 epochs are due to the change in the common stepsize. Aida has a much more compact range of layerwise average stepsizes than Adam and AdaBelief, respectively.

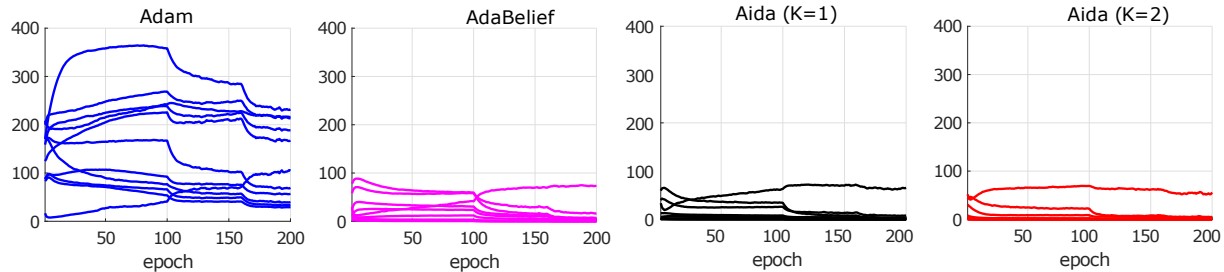

Figure 2: Comparison of layerwise standard deviations (stds) of adaptive stepsizes for the 11 neural layers by training VGG11 over CIFAR10 for 200 epochs. Aida has much smaller layerwise stds than Adam and AdaBelief, respectively.

Thirdly, extensive experimental results show that Aida with $K = 2$ yields considerably better performance than nine optimization methods for training transformer Vaswani et al. (2017) and LSTM Hochreiter & Schmidhuber (1997) models in natural language processing (NLP) tasks, and VGG11 Simonyan & Zisserman (2014) and ResNet34 in image classification tasks over CIFAR10 and CIFAR100. It is also found that Aida matches the best performance of the nine methods when training WGAN-GP models in image generation tasks. Lastly, Aida outperforms AdaBelief when training ResNet18 on the large ImageNet dataset.

**Notations**: We use small bold letters to denote vectors. The $l_2$ and $l_\infty$ norms of a vector $\boldsymbol{y}$ are denoted as $\|\boldsymbol{y}\|_2$ and $\|\boldsymbol{y}\|_\infty$, respectively. Given an $L$-layer DNN model $\boldsymbol{\theta}$ of dimension $d$, we use $\boldsymbol{\theta}_l$ of dimension $d_l$ to denote the subvector of $\boldsymbol{\theta}$ for the $l$th layer. Thus, there is $\sum_{l=1}^{L} d_l = d$. The $i$th element of $\boldsymbol{\theta}_l$ is represented by $\boldsymbol{\theta}_l[i]$. The notation $[L]$ stands for the set $[L] = \{1, \ldots, L\}$. Finally, the angle between two vectors $\boldsymbol{y}$ and $\boldsymbol{x}$ of the same dimension is denoted by $\angle \boldsymbol{xy}$.

## 2 Impact of $\epsilon$ in computation of $\mathrm{s}_t$ in AdaBelief

In this section, we study the impact of $\epsilon$ in computing the second momentum $\mathrm{s}_t$ in (3), which is missing in Zhuang et al. (2020). By inspection of (3), one can see that the parameter $\epsilon$ appears twice in the update expressions, the first one for computing $\boldsymbol{s}_t$ and the second one for computing $\boldsymbol{\theta}_t$. The impact of the second $\epsilon$ can be ignored due to the fact that $\sqrt{\epsilon/(1 - \beta_2^t)} \gg \epsilon$ when $\epsilon$ is sufficiently small (e.g., $\epsilon = 1e - 8$). As a result, we only need to focus on the first $\epsilon$ when computing $\boldsymbol{s}_t$.

Next, we show that the first $\epsilon$ in the computation of $\boldsymbol{s}_t$ helps to suppress the range of adaptive stepsizes of AdaBelief. To this purpose, we reformulate the update expressions in (3) as

$$\begin{cases} \boldsymbol{m}_t = \beta_1 \boldsymbol{m}_{t-1} + (1 - \beta_1)\boldsymbol{g}_t \\ \hat{\boldsymbol{s}}_t = \beta_2 \hat{\boldsymbol{s}}_{t-1} + (1 - \beta_2)(\boldsymbol{m}_t - \boldsymbol{g}_t)^2 \\ r_t = \beta_2 r_{t-1} + \epsilon \\ \boldsymbol{\theta}_t = \boldsymbol{\theta}_{t-1} - \eta_t \frac{1}{1-\beta_1^t} \frac{\boldsymbol{m}_t}{\sqrt{(\hat{\boldsymbol{s}}_t + r_t)/(1-\beta_2^t)}} \end{cases}, \tag{4}$$

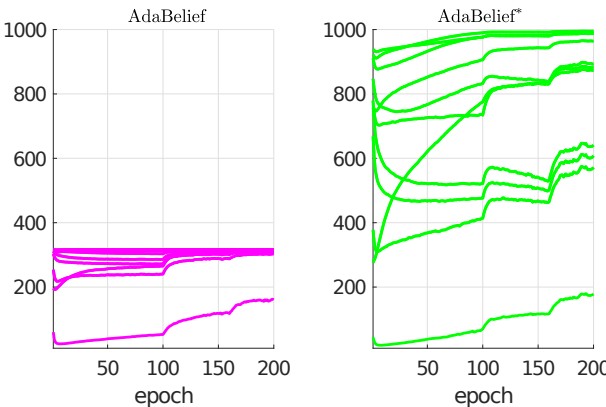

Figure 3: Comparison of layerwise average of adaptive stepsizes for the 11 neural layers by training VGG11 over CIFAR10 for 200 epochs. AdaBelief* is obtained by removing the first $\epsilon$ in the computation of $s_t$ and only keeping the second $\epsilon$. See Appendix A and C for the update procedure of AdaBelief* and the parameter setups of the two optimization methods. The optimal $\epsilon$ for AdaBelief* is selected from a discrete set that gives the best validation accuracy.

where the second $\epsilon$ is removed, and

$$s_t = \hat{s}_t + r_t = \hat{s}_t + \epsilon(1 - \beta_2^t)/(1 - \beta_2), \tag{5}$$

where $\hat{s}_0 = 0$ and $r_0 = 0$. As a result, the adaptive stepsizes $1/\sqrt{(\hat{s}_t + r_t)/(1 - \beta_2^t)}$ in (4) can be approximated as

$$1/\sqrt{(\hat{s}_t + r_t)/(1 - \beta_2^t)} = 1/\sqrt{\hat{s}_t/(1 - \beta_2^t) + \epsilon/(1 - \beta_2)} \tag{6}$$

$$\approx \frac{1}{\underbrace{\sqrt{\hat{s}_t/(1 - \beta_2^t)}}_{\text{1st term}} + \underbrace{\frac{1}{2\sqrt{\hat{s}_t/(1 - \beta_2^t)}}\epsilon/(1 - \beta_2)}_{\text{2nd term}}}, \tag{7}$$

where in the last step, the Taylor approximation is applied to a function $h(x) = \sqrt{a + x}$ around $x = 0$, where $x = \epsilon/(1 - \beta_2)$ and $a = \hat{s}_t/(1 - \beta_2^t)$.

We now investigate (7). Generally speaking, small elements of $\hat{s}_t$ lead to large adaptive stepsizes while large elements lead to small adaptive stepsizes due to the inverse operation $1/(\cdot)$. It is clear from (7) that for small elements of $\hat{s}_t$, the second term in the denominator is relatively large, implicitly penalizing large stepsizes. Furthermore, (6) indicates that those large stepsizes are upper-bounded by the quantity $1/\sqrt{\epsilon/(1 - \beta_2)}$. In contrast, for large elements of $\hat{s}_t$, the second term is relatively small, thus avoiding extremely small adaptive stepsizes. In short, including $\epsilon$ in the computation of $s_t$ suppresses the range of adaptive stepsizes in AdaBelief by avoiding extremely small stepsizes.

Fig. 3 demonstrates that when the first $\epsilon$ is removed from (3) in AdaBelief, the resulting method AdaBelief* indeed has a broader range of adaptive stepsizes than AdaBelief. At epoch 200, the eleven layerwise average stepsizes in AdaBelief* are distributed in [190,1000] while ten out of eleven layerwise average stepsizes in AdaBelief are close to a single value of 320. That is, the first $\epsilon$ in (3) indeed makes the adaptive sizes of AdaBelief more compact.

## 3 Algorithmic Design

We showed in the previous section that the particular placement of $\epsilon$ in the update expressions of AdaBelief suppresses the range of the adaptive stepsizes. In this section, we develop a new technique to further reduce the range of adaptive stepsizes of AdaBelief, which is referred to as *layerwise vector projections*. The new method is named *Aida*. A convex convergence analysis is presented at the end of the section.

$$\text{Adam: } \left\{ \boldsymbol{g}_{l,t}^2 | t \geq 0 \right\}$$

$$\text{AdaBelief: } \left\{ (\boldsymbol{m}_{l,t} - \boldsymbol{g}_{l,t})^2 | t \geq 0 \right\}$$

$$\text{Aida: } \left\{ (\boldsymbol{m}_{l,t}^{(K)} - \boldsymbol{g}_{l,t}^{(K)})^2 | t \geq 0 \right\}$$

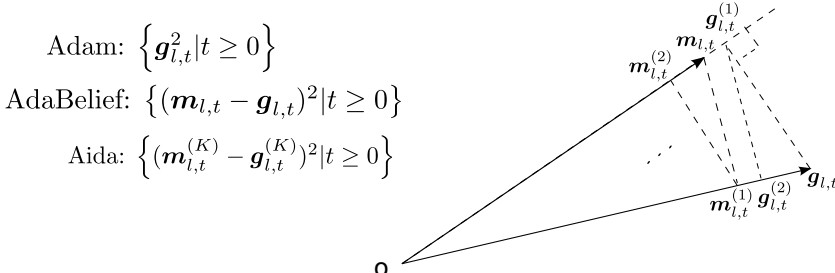

Figure 4: Computation of $\{(\boldsymbol{m}_{l,t}^{(k)}, \boldsymbol{g}_{l,t}^{(k)}) | k = [K]\}$ by starting from the pair $(\boldsymbol{m}_{l,t}, \boldsymbol{g}_{l,t})$ via sequential and alternating vector projections in Aida.

### 3.1 Motivation

Our aim is to design a new adaptive optimization algorithm, in which the range of the adaptive stepsizes is smaller than that of AdaBelief. To achieve the above goal, we consider processing $\boldsymbol{m}_t$ and $\boldsymbol{g}_t$ in a layerwise manner at iteration $t$ before using them to estimate the second momentum $\boldsymbol{v}_t$. Due to the nature of back-propagation when training a DNN model $\boldsymbol{\theta}$, it is computationally more efficient to perform layerwise processing than operating on the entire vector $\boldsymbol{m}_t$ and $\boldsymbol{g}_t$.

Intuitively speaking, when the variance of the adaptive stepsizes within a particular neural layer is encouraged to be small by layerwise manipulation, the update for the model parameters within the same neural layer would become relatively robust to elementwise gradient outliers (exploding or vanishing gradient elements across iterations). The recent work You et al. (2020) proposed the LARS and LAMB optimizers. They are, respectively, extensions of SGD with momentum and Adam that introduce layerwise normalisation in the update of a DNN model per iteration. The purpose for introducing the layerwise normalization in You et al. (2020) is to provide robustness to exploding layerwise gradients and plateaus for the scenario of utilizing extremely large batchsizes to train large DNN models. Differently from You et al. (2020), our work considers the scenario of normal training batchsizes.

We now study what kind of layerwise operation is desirable for reducing the layerwise variance of the adaptive stepsizes in AdaBelief. Firstly, we note that the parameter $\epsilon$ of (3) in AdaBelief essentially defines an upper bound on the adaptive stepsizes and is independent of neural layer and iteration indices. By inspection of (4)-(6), the upper bound can be expressed as

$$\frac{1}{\sqrt{r_t/(1 - \beta_2^t)} + \epsilon} = \frac{1}{\sqrt{\epsilon/(1 - \beta_2)} + \epsilon}. \tag{8}$$

We use $(\boldsymbol{m}_{l,t}, \boldsymbol{g}_{l,t})$ to denote the subvectors of $(\boldsymbol{m}_t, \boldsymbol{g}_t)$ for the $l$th neural layer. Suppose we track the EMA of $(\gamma_{l,t} \boldsymbol{m}_{l,t} - \beta_{l,t} \boldsymbol{g}_{l,t})^2$ for the $l$th layer, where $0 < \gamma_{l,t}, \beta_{l,t} \leq 1$ are functions of $(\boldsymbol{m}_{l,t}, \boldsymbol{g}_{l,t})$, instead of the EMA of $(\boldsymbol{m}_{l,t} - \boldsymbol{g}_{l,t})^2$ being tracked in AdaBelief. If the scalars $\{\gamma_{l,t}, \beta_{l,t}\}$ are sufficiently small in the extreme case, all the adaptive stepsizes of the new method tend to approach the upper bound in (8). As a result, the new method will have a smaller range of adaptive stepsizes than AdaBelief either in a layerwise manner or globally.

We propose to compute the scalars $(\gamma_{l,t}, \beta_{l,t})$ of $\boldsymbol{m}_{l,t}$ and $\boldsymbol{g}_{l,t}$ mentioned above via $K$ mutual-vector projections starting from $(\boldsymbol{m}_{l,t}, \boldsymbol{g}_{l,t})$ (see Fig. 4 for demonstration). In practice, it is found that $K = 2$ is sufficient to produce small scalars $(\gamma_{l,t}, \beta_{l,t})$, leading to a smaller range of adaptive stepsizes than those of AdaBelief and Adam. The parameter $K$ of Aida in Fig. 1-2 was set to $K \in \{1, 2\}$. In the following, the update expressions of Aida are presented in detail.

### 3.2 Aida as an extension of AdaBelief

Consider the $l$th layer of a DNN model at iteration $t$. We perform a sequence of mutual-vector projections to obtain a set of projected vectors $\{(\boldsymbol{m}_{l,t}^{(k)}, \boldsymbol{g}_{l,t}^{(k)}) | k = [K]\}$ starting from the initial pair $(\boldsymbol{m}_{l,t}^{(0)}, \boldsymbol{g}_{l,t}^{(0)}) = (\boldsymbol{m}_{l,t}, \boldsymbol{g}_{l,t})$. Using

algebra, the two vectors at iteration $k$ can be represented as

$$\boldsymbol{m}_{l,t}^{(k+1)} = \frac{\langle \boldsymbol{g}_{l,t}^{(k)}, \boldsymbol{m}_{l,t}^{(k)} \rangle}{\|\boldsymbol{g}_{l,t}^{k}\|_2^2 + \xi} \boldsymbol{g}_{l,t}^{(k)} \tag{9}$$

$$\boldsymbol{g}_{l,t}^{(k+1)} = \frac{\langle \boldsymbol{g}_{l,t}^{(k)}, \boldsymbol{m}_{l,t}^{(k)} \rangle}{\|\boldsymbol{m}_{l,t}^{(k)}\|_2^2 + \xi} \boldsymbol{m}_{l,t}^{(k)}, \tag{10}$$

where $\langle \cdot, \cdot \rangle$ denotes the inner product, and $\xi > 0$ is a scalar parameter to make sure the division operations are valid. The above two projections (9)-(10) ensure that the resulting projected vectors share the same vector-direction as either $\boldsymbol{m}_{l,t}$ or $\boldsymbol{g}_{l,t}$. See Fig. 4 for visualisation.

**Remark 1.** *We note that the $K$ mutual-vector projections starting from the pair of $\boldsymbol{g}_{l,t}$ and $\boldsymbol{m}_{l,t}$ can, in principle, be realized by scaling the two vectors via the $K$th power of the cosine of $\angle \boldsymbol{g}_{l,t} \boldsymbol{m}_{l,t}$, which would make the additional computational overhead roughly constant as a function of $K$. In practice, we found that a positive value of the parameter $\xi$ in (9) and (10) of the mutual-vector projections improves the generalization performance, which, for simplicity, was set to $1e - 20$ across all the experiments in the paper. We refer the reader to Appendix E for supplementary experiments on the utilization of the $K$th power of the cosine of $\angle \boldsymbol{g}_{l,t} \boldsymbol{m}_{l,t}$ to perform the scaling. The empirical study in Subsection 4.3 indicates that Aida with $K = 2$ only introduces a small computational overhead compared to AdaBelief, which makes the formulation of the mutual-vector projections (9) and (10) practically affordable.*

Once $(\boldsymbol{m}_{l,t}^{(K)}, \boldsymbol{g}_{l,t}^{(K)})$ are obtained for the $l$th layer, Aida tracks the EMA of the squared difference $(\boldsymbol{m}_{l,t}^{(K)} - \boldsymbol{g}_{l,t}^{K})^2$, given by

$$\boldsymbol{v}_{l,t} = \beta_2 \boldsymbol{v}_{l,t-1} + (1 - \beta_2) \left( \boldsymbol{m}_{l,t}^{(K)} - \boldsymbol{g}_{l,t}^{(K)} \right)^2 + \epsilon, \tag{11}$$

where $1 > \beta_2 > 0$, and $\epsilon > 0$ is added as recommended by our earlier analysis. With $\boldsymbol{v}_{l,t}$, the model parameters $\boldsymbol{\theta}_{l,t}$ of the $l$th layer can be updated accordingly. See Algorithm 1 for a summary of Aida.

Next, we consider the geometric properties of the set of projected vectors. It is not difficult to show that after projection, the resulting vectors have either shorter or equal length in comparison to the original vectors:

$$\|\boldsymbol{m}_{l,t}^{(k)}\|_2 \leq \|\boldsymbol{m}_{l,t}^{k-1}\|_2 \quad \text{and} \quad \|\boldsymbol{g}_{l,t}^{(k)}\|_2 \leq \|\boldsymbol{g}_{l,t}^{(k-1)}\|_2. \tag{12}$$

Using the fact that mutual projections of two vectors do not change the angle, we then have

$$\|\boldsymbol{m}_{l,t}^{(k)} - \boldsymbol{g}_{l,t}^{(k)}\|_2 \leq \|\boldsymbol{m}_{l,t}^{(k-1)} - \boldsymbol{g}_{l,t}^{(k-1)}\|_2, \tag{13}$$

where the equality holds if $\boldsymbol{m}_{l,t}$ and $\boldsymbol{g}_{l,t}$ are on the same line and $\xi$ can be ignored in (10).

For the extreme case that each neural layer has only one parameter (i.e., $\boldsymbol{g}_{l,t} \in \mathbb{R}, \forall l \in [L]$), it is easy to show that the projection operation has no effect. That is, $(\boldsymbol{g}_{l,t}^{(k)} - \boldsymbol{m}_{l,t}^{(k)})^2 = (\boldsymbol{g}_{l,t} - \boldsymbol{m}_{l,t})^2$ for all $k \in [K]$ if $\xi$ is ignored in (10). In this case, Aida reduces to AdaBelief.

From the above analysis, we can conclude that the EMA of $(\boldsymbol{m}_{l,t}^{(K)} - \boldsymbol{g}_{l,t}^{K})^2$ for the $l$th layer can be viewed as the EMA of $(\gamma_{K,l,t} \boldsymbol{m}_{l,t} - \beta_{K,l,t} \boldsymbol{g}_{l,t})^2$, where the scalars $\gamma_{K,l,t}, \beta_{K,l,t} \in (0, 1]$. In general, the angles $\{\angle \boldsymbol{g}_{l,t} \boldsymbol{m}_{l,t} | t \geq 0\}$ would be non-zero due to randomness introduced by the minibatch training strategy in a typical DNN task. As a result, increasing the number $K$ of vector projections would cause the elements of $\{\gamma_{K,l,t}, \beta_{K,l,t} | t \geq 0\}$ to approach zero. In other words, the parameter $K$ controls the range of the adaptive stepsizes of Aida. A larger $K$ makes the adaptive stepsizes more compact.

Figs. 1 and 2 provide empirical evidence that Aida does indeed have a smaller range of adaptive stepsizes than AdaBelief and Adam. Furthermore, as $K$ increases from 1 to 2, the range of adaptive stepsizes of Aida becomes increasingly compact. Hence, Aida makes a bridge between SGD with momentum and AdaBelief. As will be demonstrated in the experiments, Aida improves the generalization of Adam and AdaBelief for several classical DNN tasks.

---

**Algorithm 1** Aida: Suppressing the range of adaptive stepsizes of AdaBelief by layerwise vector projections

---

**Input:** $\beta_1, \beta_2, \eta_t, \epsilon > 0, \xi = 1e - 20, K = 2$
**Init.:** $\boldsymbol{\theta}_0 \in \mathbb{R}^d, \boldsymbol{m}_0 = 0, \boldsymbol{v}_0 = \tilde{\boldsymbol{v}}_0 = 0 \in \mathbb{R}^d$
**for** $t = 1, 2, \ldots, T$ **do**
   $\boldsymbol{g}_t \leftarrow \nabla f(\boldsymbol{\theta}_{t-1})$
   $\boldsymbol{m}_t \leftarrow \beta_1 \boldsymbol{m}_{t-1} + (1 - \beta_1)\boldsymbol{g}_t$
   **for** $l = 1, \ldots, L$ **do**
     $\boldsymbol{m}_{l,t}^{(0)} = \boldsymbol{m}_t^{(0)}, \boldsymbol{g}_{l,t}^{(0)} = \boldsymbol{g}_t$
     **for** $k = 1, \ldots, K$ **do**
       $\boldsymbol{m}_{l,t}^{(k)} = \frac{\langle \boldsymbol{m}_{l,t}^{k-1}, \boldsymbol{g}_{l,t}^{k-1} \rangle}{\|\boldsymbol{g}_{l,t}^{(k-1)}\|_2^2 + \xi} \boldsymbol{g}_{l,t}^{(k-1)}$
       $\boldsymbol{g}_{l,t}^{(k)} = \frac{\langle \boldsymbol{m}_{l,t}^{k-1}, \boldsymbol{g}_{l,t}^{k-1} \rangle}{\|\boldsymbol{m}_{l,t}^{(k-1)}\|_2^2 + \xi} \boldsymbol{m}_{l,t}^{(k-1)}$
     **end for**
     $\boldsymbol{v}_{l,t} \leftarrow \beta_2 \boldsymbol{v}_{l,t-1} + (1 - \beta_2)(\boldsymbol{m}_{l,t}^{(K)} - \boldsymbol{g}_{l,t}^{(K)})^2 + \epsilon$
   **end for**
   $\tilde{\boldsymbol{m}}_t \leftarrow \frac{\boldsymbol{m}_t}{1 - \beta_1^t} \quad \left\{ \tilde{\boldsymbol{v}}_{l,t} \leftarrow \frac{\boldsymbol{v}_{l,t}}{1 - \beta_2^t} \right\}_{l=1}^{L}$
   $\boldsymbol{\theta}_t \leftarrow \boldsymbol{\theta}_{t-1} - \frac{\eta_t \tilde{\boldsymbol{m}}_t}{\sqrt{\tilde{\boldsymbol{v}}_t}}$
**end for**
**Output:** $\boldsymbol{\theta}_T$

---

## 3.3 Convergence analysis

In this paper, we study the convergence of Aida for convex optimization. Our analysis follows a strategy similar to that used to analyse AdaBelief in Zhuang et al. (2020).

**Theorem 1.** *Suppose $\{\boldsymbol{\theta}_t\}_{t=0}^{T}$ and $\{\boldsymbol{v}_t\}_{t=0}^{T}$ are the iterative updates obtained by either Aida[4] starting with $(\boldsymbol{m}_0, \boldsymbol{v}_0) = (\boldsymbol{0}, \boldsymbol{0})$. Let $0 \leq \beta_{1t} = \beta_1 \lambda^t < 1, 0 \leq \beta_2 < 1$, and $\eta_t = \frac{\eta}{\sqrt{t}}$. Assume (1): $f(\boldsymbol{\theta})$ is a differentiable convex function with $\|\boldsymbol{g}_t\|_\infty \leq G_\infty/2$ (hence $\|\boldsymbol{m}_{l,t}^{(K)} - \boldsymbol{g}_{l,t}^{(K)}\|_\infty \leq G_\infty$) for all $t \in [T]$; (2): the updates $\{\boldsymbol{\theta}_t\}_{t=0}^{T}$ and the optimal solution $\boldsymbol{\theta}^*$ are bounded by a hyper-sphere, i.e., $\|\boldsymbol{\theta}_t\|_2 \leq D$ and $\|\boldsymbol{\theta}^*\|_2 \leq D$; (3): $0 < c \leq \tilde{\boldsymbol{v}}_t[i] \leq \tilde{\boldsymbol{v}}_{t-1}[i]$ for all $i \in \{1, \ldots, d\}$ and $t \in [T]$. Denote $\bar{\boldsymbol{\theta}}_T = \frac{1}{T} \sum_{t=0}^{T-1} \boldsymbol{\theta}_t$ and $\boldsymbol{g}_{1:T}^2[i] = ((\boldsymbol{g}_1[i])^2, \ldots, (\boldsymbol{g}_T[i])^2) \in \mathbb{R}^T$. We then have the following bound on regret:*

$$f(\bar{\boldsymbol{\theta}}_T) - f(\boldsymbol{\theta}^*) \leq \frac{D^2 d(G_\infty + \sqrt{\epsilon})}{\eta(1 - \beta_1)(1 - \beta_2)T} + \frac{D^2 d(G_\infty + \sqrt{\epsilon})}{\sqrt{T}\eta(1 - \beta_1)(1 - \beta_2)} + \frac{(1 + \beta_1)}{2(1 - \beta_1)T} \frac{\eta\sqrt{1 + \log T}}{\sqrt{c}(1 - \beta_1)^2} \|(\boldsymbol{g}_{1:T}^2[i])\|_2$$
$$+ \frac{D^2 \beta_1 d(G_\infty + \sqrt{\epsilon})}{T(1 - \beta_1)(1 - \beta_2)\eta(1 - \lambda)^2}. \tag{14}$$

*Proof.* See Appendix B for proof. □

We note that the major difference between our analysis for Aida and that for AdaBelief in Zhuang et al. (2020) is that we assume $\tilde{\boldsymbol{v}}_t[i] \leq \tilde{\boldsymbol{v}}_t[i-1]$ while the analysis in Zhuang et al. (2020) essentially utilizes the condition $\tilde{\boldsymbol{v}}_t[i] \geq \tilde{\boldsymbol{v}}_t[i-1]$ for convergence analysis. Our motivation for the new condition is that as the iteration index $t$ increases, the gradient $\boldsymbol{g}_t$ tends to approach zero, which makes $\tilde{\boldsymbol{v}}_t$ approach to zero.

# 4 Experiments

We evaluated Aida on three types of DNN tasks: (1) natural language processing (NLP) on training transformer and LSTM models; (2) image classification on training VGG and ResNet He et al. (2015) models; (3) image generation

---

[4]$\beta_1$ in Algorithm 1 is generalized to be $\beta_{1t}, t \geq 0$ to facilitate convergence analysis.

on training WGAN-GP Gulrajani et al. (2017). Two open-source repositories were used for the above DNN training tasks. The first repository[5] was adopted for the task of training a transformer. The second one[6] was used for all the remaining tasks, which includes the original implementation of AdaBelief. The second repository was used to compare AdaBelief with SGD with momentum and seven adaptive optimization algorithms from the literature, namely Yogi Zaheer et al. (2018), RAdam Liu et al. (2019), MSVAG Balles & Hennig (2017), Fromage Bernstein et al. (2020), Adam Kingma & Ba (2017), AdaBound Luo et al. (2019), and AdamW Loshchilov & Hutter (2019). We compared our Aida algorithm with all the above optimization methods.

We now briefly explain the selection of hyper-parameters for the tested optimizers, of which the detailed information was provided in Appendix D. For our experiments on using the second repository, the tested hyper-parameters were inherited from the source code for AdaBelief. To be specific, in the source code of AdaBelief, the optimal parameter $\epsilon$ of AdaBelief was, in general, set differently for different DNN tasks. The setting for the parameter $\epsilon$ in Aida follows that of the Adabelief setup in most cases and hence also was configured differently for different tasks. To make a fair comparison with Aida and AdaBelief, the optimal parameter $\epsilon$ was searched for the other adaptive optimizers for each DNN task. For each optimizer, we stopped searching when its performance dropped significantly, indicating the value of $\epsilon$ was either too large or too small. Lastly, when training LSTM models in the open-source of AdaBelief, the initial learning rate $\eta_0$ of the tested optimizers was searched together with $\epsilon$. Consequently, in our work, we also searched the initial learning rate for those optimizers when training LSTMs.

It was found that Aida with $K = 2$ outperforms the nine reference methods for training transformer, LSTM, VGG11, and ResNet34 models while it matches the best performance of the nine methods for training WGAN-GP. Furthermore, experiments on training ResNet18 on the large ImageNet dataset show that Aida outperforms AdaBelief.

The time complexity of Aida was evaluated for training VGG11 and ResNet34 on a 2080 Ti GPU. In brief, Aida with $K = 2$ consumed $25\%$ more time per epoch compared to AdaBelief.

## 4.1 On training a transformer

In this task, we consider the training of a transformer for WMT16: multimodal translation by using the first open-source as indicated in the footnote. In the training process, we retained almost all of the default hyper-parameters provided in the open-source except for the batch size. Due to limited GPU memory, we changed the batch size from 256 to 200. The parameters of Aida were set to $(\eta_0, \beta_1, \beta_2, \epsilon) = (0.001, 0.9, 0.98, 1e{-}16)$. The parameter-setups for other optimizers can be found in Table 7 of Appendix D, where certain hyper-parameters for each optimizer were searched over some discrete sets to optimize the validation performance. For example, the parameter $\epsilon$ of Adam was searched over the set $\{1e{-}6, 1e{-}7, \ldots, 1e{-}16\}$ while the remaining parameters were set to $(\eta_0, \beta_1, \beta_2) = (0.001, 0.9, 0.98)$ as in Aida. Once the optimal parameter-configuration for each optimizer was obtained by searching, three experimental repetitions were then performed to alleviate the effect of the randomness.

It is clear from Table 1 and Fig. 5 that Aida significantly outperforms all other methods. We emphasize that the maximum number of epochs was set to 400 for each optimizer by following the default setup in the open source, and no epoch cutoff is performed in favor of Aida. When $K$ increases from 1 to 2 in Aida, the method converges considerably faster and produces better validation performance, which may be due to the fact that Aida with $K = 2$ has a more compact range of adaptive stepsizes. On the other hand, the non-adaptive method SGD with momentum produces a performance that is inferior to all adaptive methods except Fromage and MSVAG.

## 4.2 On training LSTMs

In this experiment, we consider training LSTMs with a different number of layers over the Penn TreeBank dataset Marcus et al. (1993) by utilizing the second open-source repository for AdaBelief. As explained earlier, when training LSTMs in the original source code, both the initial learning rate $\eta_0$ and the parameter $\epsilon$ were searched in those adaptive optimizers. In this work, we followed a similar procedure. See Table 8 in Appendix D for a summary of the fixed and free parameters for each optimizer. An example is Adam for which $\eta_0 \in \{0.01, 0.001\}$ and $\epsilon \in \{1e{-}6, 1e{-}7, \ldots, 1e{-}16\}$ were tested to find the optimal configuration that produces the best validation performance.

---

[5] https://github.com/jadore801120/attention-is-all-you-need-pytorch
[6] https://github.com/juntang-zhuang/Adabelief-Optimizer

Table 1: Performance comparison for training the transformer

| SGD (non-adaptive) | 55.58±0.34 | | |
|---|---|---|---|
| AdaBound | 55.90±0.21 | Yogi | 60.47±0.61 |
| RAdam | 64.47±0.19 | MSVAG | 53.79±0.13 |
| Fromage | 35.57±0.19 | Adam | 64.71±0.57 |
| AdamW | 64.49±0.24 | AdaBelief | 66.90±0.77 |
| Aida(K=1) | 68.77±0.16 | Aida(K=2) | **68.96**±0.06 |

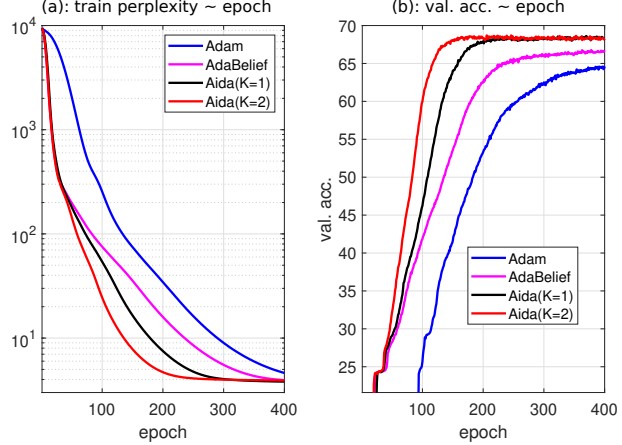

Figure 5: Performance visualisation of Aida, Adam, and AdaBelief for the training of the transformer. See Fig. 7 in Appendix F for the plots of the associated learning curves versus runtime.

Table 2: Validation perplexity on Penn Treebank for 1, 2, 3-layer LSTM. **lower** is better.

| | Aida($K$=1) | AdaBelief | AdamW | Adam | Yogi | AdaBound |
|---|---|---|---|---|---|---|
| 1 layer | 82.27 | 84.21 | 88.36 | 84.28 | 86.78 | 84.52 |
| 2 layer | 66.16 | 66.29 | 73.18 | 66.86 | 71.56 | 67.01 |
| 3 layer | 61.98 | 61.23 | 70.08 | 64.28 | 67.83 | 63.16 |
| | Aida(K=2) | SGD (non-adaptive) | RAdam | MSVAG | Fromage | |
| 1 layer | **81.53** | 85.52 | 88.76 | 84.75 | 85.20 | |
| 2 layer | **65.04** | 67.44 | 74.12 | 68.91 | 72.22 | |
| 3 layer | **60.18** | 63.68 | 70.41 | 65.04 | 67.37 | |

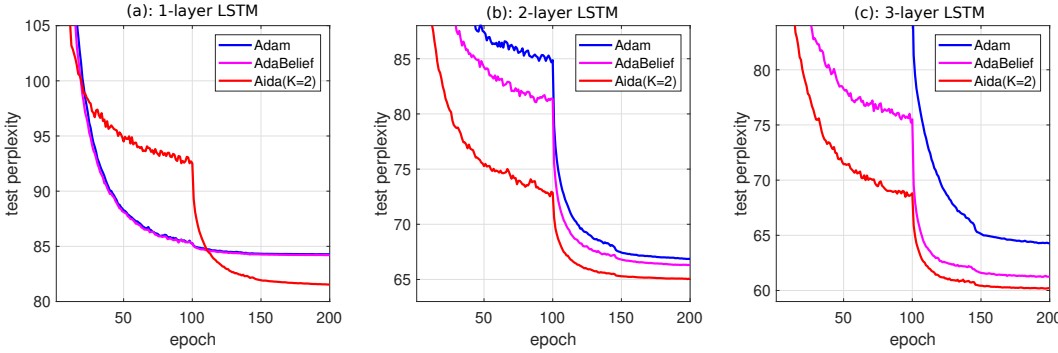

Figure 6: Performance visualisation of Aida, AdaBelief, and Adam in Table 2. See Fig. 8 in Appendix F for the plots of the associated learning curves versus runtime.

Table 3: Validation accuracies (in percentage) and time complexity in seconds per epoch (referred to as t. c.) of nine methods for training VGG11 and ResNet34 over CIFAR10 and CIFAR100. The best result is highlighted in each column.

| optimizers | CIFAR10 | | | | CIFAR100 | | | |
| | VGG11 | | ResNet34 | | VGG11 | | ResNet34 | |
| | val. acc | t. c. | val. acc | t. c. | val. acc | t. c. | val. acc | t. c. |
|---|---|---|---|---|---|---|---|---|
| SGD (non-adaptive) | 91.36±0.07 | **5.83** | 95.48±0.11 | **30.45** | 67.02±0.25 | **5.85** | 78.10±0.18 | **30.92** |
| Yogi | 90.74±0.16 | 6.49 | 94.98±0.26 | 31.74 | 65.57±0.17 | 6.42 | 77.17±0.12 | 32.20 |
| RAdam | 89.58±0.10 | 6.28 | 94.64±0.18 | 31.21 | 63.62±0.20 | 6.29 | 74.87±0.13 | 31.58 |
| MSVAG | 90.04±0.22 | 7.08 | 94.65±0.08 | 33.78 | 62.67±0.33 | 7.19 | 75.57±0.14 | 33.80 |
| Fromage | 89.72±0.25 | 6.66 | 94.64±0.07 | 35.19 | 62.93±0.53 | 6.56 | 74.84±0.27 | 35.50 |
| Adam | 91.20±0.21 | 6.15 | 95.09±0.18 | 31.28 | 67.88±0.13 | 6.20 | 77.31±0.14 | 31.47 |
| AdamW | 89.46±0.08 | 6.25 | 94.48±0.18 | 31.71 | 62.50±0.23 | 6.31 | 74.29±0.20 | 31.80 |
| AdaBound | 90.48±0.12 | 6.71 | 94.73±0.16 | 33.75 | 64.80±0.42 | 6.73 | 76.15±0.10 | 33.78 |
| AdaBelief | 91.55±0.13 | 6.47 | 95.15±0.11 | 31.66 | 68.05±0.31 | 6.49 | 77.32±0.37 | 31.74 |
| Aida($K$=1) | 91.52±0.05 | 7.27 | 95.31±0.05 | 35.25 | 68.89±0.09 | 7.32 | 77.50±0.12 | 35.46 |
| Aida($K$=2) | **91.68**±0.16 | 7.95 | **95.57**±0.13 | 39.53 | **69.02**±0.11 | 8.01 | **78.86**±0.12 | 39.64 |

Table 4: Best FID obtained for each optimizer (**lower** is better)

| | Aida($K$=2) | Aida($K$=1) | AdaBelief | Adam | RAdam | AdaBound |
|---|---|---|---|---|---|---|
| best FIDs | 55.7 | **55.65** | 56.73 | 66.71 | 69.14 | 61.65 |
| | AdamW | MSVAG | SGD | Yogi | Fromage | |
| best FIDs | 63.76 | 69.47 | 90.61 | 68.34 | 78.47 | |

We emphasize that the parameters of Aida were set to $(\eta_0, \beta_1, \beta_2, \epsilon) = (0.001, 0.9, 0.999, 1e-16)$, whereas $\eta_0$ and $\epsilon$ remain the same for all different numbers of neural layers in LSTMs.

Table. 2 summarises the obtained validation perplexities of the ten methods for training 1, 2, and 3-layer LSTMs. It was found that for each optimizer, independent experimental repetitions lead to almost the same validation perplexity value. Therefore, we only keep the average of the validation perplexity values from three independent experimental repetitions for each experimental setup in the table and ignore the standard deviations.

It is clear from Table. 2 that Aida again outperforms all other methods in all three scenarios. Fig. 6 further visualises the validation performance of Aida compared to AdaBelief and Adam. The performance gain of Aida is considerable in all three scenarios. One observes that in the initial training stage, Aida converges slower than AdaBelief only for the 1-layer LSTM. This is because the optimal parameters $(\eta_0, \epsilon)$ in AdaBelief are different for different numbers of neural layers. In particular, the optimal setups in AdaBelief are: $(\eta_0, \epsilon) = (0.001, 1e-16)$ for 1-layer LSTM, and $(\eta_0, \epsilon) = (0.01, 1e-12)$ for 2-layer and 3-layer LSTMs. When the parameters of AdaBelief for the 2-layer and 3-layer LSTMs are also set to $(\eta_0, \epsilon) = (0.001, 1e-16)$ to be in line with the setup of Aida, it is found that Aida also converges slower than AdaBelief in the beginning of the training process.

### 4.3 On training VGG11 and ResNet34 over CIFAR10 and CIFAR100

In this task, the ten optimizers were evaluated by following a similar experimental setup as in Zhuang et al. (2020). In particular, the batch size and epoch were set to 128 and 200, respectively. The common stepsize $\eta_t$ is reduced by multiplying by 0.1 at 100 and 160 epochs. The detailed parameter-setups for the optimizers can be found in Table 10 in Appendix D. Three experimental repetitions were conducted for each optimizer to alleviate the effect of randomness.

Both the validation performance and the algorithmic complexity are summarised in Table 3. It is clear that Aida with $K = 2$ consistently outperforms the nine reference methods in terms of validation accuracies at the cost of additional

computational time. This demonstrates that the compact range of adaptive stepsizes in Aida does indeed improve the generalization performance.

We can also conclude from the table that SGD with momentum is the most computationally efficient method. On the other hand, due to the layerwise vector projections, Aida with $K = 2$ consumed an additional $25\%$ time per epoch compared to AdaBelief.

### 4.4 On training WGAN-GP over CIFAR10

This task focuses on training WGAN-GP, where the setup $\beta_1 = 0.5$ follows the original setting in training AdaBelief. The parameters of Aida were set to $(\eta_t, \beta_1, \beta_2, \epsilon) = (0.0002, 0.5, 0.999, 1e-12)$, in agreement with the recommended setup of AdaBelief in the original open-source. The other eight optimizers have both fixed and free parameters, details of which can be found in Table 9 of Appendix D. As an example, the free parameter of Adam is $\epsilon \in \{1e-4, 1e-6, \ldots, 1e-14\}$. For each parameter-configuration of an optimizer, three experimental repetitions were performed due to the relatively unstable Frechet inception distance (FID) scores in training WGAN-GP.

Table 4 shows the best FID for each method. It can be seen from the table that Aida with $K \in \{1, 2\}$ provides better performance than AdaBelief, while the other methods perform significantly worse. It is worth noting that the FID for AdaBelief is better than the reported results in Zhuang et al. (2020) which may be due to different versions of python packages.

### 4.5 On training ResNet18 over ImageNet

In the last experiment, we investigated the performance gain of Aida compared to AdaBelief for training ResNet18 on the large ImageNet dataset. The maximum epoch and minibatch size were set to 90 and 256, respectively. The common stepsize $\eta_t$ is dropped by a factor of 0.1 at 70 and 80 epochs. The parameter setup for the two optimizers can be found in Table 6 of Appendix D. To make a fair comparison, the parameter $\epsilon$ of AdaBelief was searched over the discrete set $\{1e-8, 1e-9, 1e-10\}$ instead of using the recommended setup of $\epsilon = 1e-8$ in the original repository for AdaBelief. Three experimental repetitions were conducted for each optimizer to mitigate the effect of randomness.

It is seen from Table 5 that for the large ImageNet dataset, Aida performs slightly better than AdaBelief, indicating that the performance gain of Aida is robust against different sizes of datasets. We note that the performance of AdaBelief is slightly lower than the reported result in Zhuang et al. (2020), where no standard error is reported. This might be either because we repeated the experiments three times per optimizer or due to the fact that different versions of the Python packages were used, which are difficult to track.

From an overall perspective, all five experiments in our paper demonstrate that Aida consistently outperforms AdaBelief. The performance gain is significant for certain tasks and small for other tasks. Our results suggest that proper manipulation of the range of adaptive stepsizes of an adaptive optimizer improves generalization performance. We hope our work will lead to the development of related techniques to improve the performance of other existing optimizers.

Table 5: Validation accuracies (in percentage) of AdaBelief and Aida for training ResNet18 over ImageNet.

| optimizers | AdaBelief | Aida($K = 2$) |
|---|---|---|
| val. acc. | 69.65$\pm$0.06 | **69.70**$\pm$0.08 |

## 5 Conclusions

In this paper, we have shown that the range of the adaptive stepsizes of DNN optimizers has a significant impact on performance. The proposed Aida optimizer suppresses the range of the adaptive stepsizes of AdaBelief making it closer to SGD with momentum. Our experimental results indicate that Aida will be able to produce better performance across a wide range of DNN-based applications.

In the design of the Aida optimizer, we track the EMA (or equivalently the second momentum) of $(\gamma_{l,t} \boldsymbol{m}_{l,t} - \beta_{l,t} \boldsymbol{g}_{l,t})^2$ for the $l$th layer of a DNN model as opposed to $(\boldsymbol{m}_{l,t} - \boldsymbol{g}_{l,t})^2$ used in AdaBelief, where $\gamma_{l,t}, \beta_{l,t} \in (0, 1]$ are obtained by vector projections. Consequently, the adaptive stepsizes of Aida have a more compact range than those of AdaBelief.

Our empirical study shows that Aida with $K = 2$ outperforms nine optimizers including Adam and AdaBelief for training transformer, LSTM, VGG11, and ResNet34 models while at the same time, it matches the best performance of the nine methods for training WGAN-GP models. In addition, experiments on training ResNet18 over the large ImageNet dataset show that Aida performs better than AdaBelief. On the other hand, it was found that the non-adaptive method SGD with momentum only produces good performance when training VGG and ResNet models. This suggests that the *adaptivity* of Aida is important, allowing the method to effectively train different types of DNN models.

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

## A    Update procedure of AdaBelief*

The first $\epsilon$ is removed in (3) to verify if AdaBelief* has a broad range of adaptive stepsizes than AdaBelief.

---

**Algorithm 2** AdaBelief*

---
1: **Input:** $\beta_1, \beta_2, \eta_t, \epsilon > 0$
2: **Init.:** $\boldsymbol{\theta}_0 \in \mathbb{R}^d$, $\boldsymbol{m}_0 = 0$, $\boldsymbol{v}_0 = 0 \in \mathbb{R}^d$
3: **for** $t = 1, 2, \ldots, T$ **do**
4:     $\boldsymbol{g}_t \leftarrow \nabla f(\boldsymbol{\theta}_{t-1})$
5:     $\boldsymbol{m}_t \leftarrow \beta_1 \boldsymbol{m}_{t-1} + (1 - \beta_1)\boldsymbol{g}_t$
6:     $q_t \leftarrow \beta_2 q_{t-1} + (1 - \beta_2)(\boldsymbol{m}_t - \boldsymbol{g}_t)^2$
7:     $\tilde{\boldsymbol{m}}_t \leftarrow \frac{\boldsymbol{m}_t}{1 - \beta_1^t}$   $\tilde{q}_t \leftarrow \frac{q_t}{1 - \beta_2^t}$
8:     $\boldsymbol{\theta}_t \leftarrow \boldsymbol{\theta}_{t-1} - \frac{\eta_t}{\sqrt{\tilde{q}_t} + \epsilon}\tilde{\boldsymbol{m}}_t$
9: **end for**
10: **Output:** $\boldsymbol{\theta}_T$

---

## B    Convex Convergence Analysis

Firstly, we note that the bias term $1 - \beta_1^t$ in the update expressions of Aida in Algorithm 1 can be absorbed into the common stepsize $\eta_t$. Therefore, we will ignore the bias term in the following proof. Suppose $\boldsymbol{\theta}^*$ is the optimal solution for solving the convex optimization problem, i.e., $\boldsymbol{\theta}^* = \arg\min_{\boldsymbol{\theta}_t} f(\boldsymbol{\theta})$. Using the fact that $\boldsymbol{\theta}_t = \boldsymbol{\theta}_{t-1} - \eta_t \tilde{\boldsymbol{v}}_t^{-1/2} \boldsymbol{m}_t$, we have

$$
\begin{aligned}
&\|\tilde{\boldsymbol{v}}_t^{1/4}(\boldsymbol{\theta}_t - \boldsymbol{\theta}^*)\|_2^2 \\
&= \|\tilde{\boldsymbol{v}}_t^{1/4}(\boldsymbol{\theta}_{t-1} - \eta_t \tilde{\boldsymbol{v}}_t^{-1/2} \boldsymbol{m}_t - \boldsymbol{\theta}^*)\|_2^2 \\
&= \|\tilde{\boldsymbol{v}}_t^{1/4}(\boldsymbol{\theta}_{t-1} - \boldsymbol{\theta}^*)\|_2^2 + \eta_t^2 \|\tilde{\boldsymbol{v}}_t^{-1/4} \boldsymbol{m}_t\|_2^2 - 2\eta_t \langle \beta_{1t} \boldsymbol{m}_{t-1} + (1 - \beta_{1t})\boldsymbol{g}_t, \boldsymbol{\theta}_{t-1} - \boldsymbol{\theta}^* \rangle \\
&= \|\tilde{\boldsymbol{v}}_t^{1/4}(\boldsymbol{\theta}_{t-1} - \boldsymbol{\theta}^*)\|_2^2 + \eta_t^2 \|\tilde{\boldsymbol{v}}_t^{-1/4} \boldsymbol{m}_t\|_2^2 - 2\eta_t(1 - \beta_{1t})\langle \boldsymbol{g}_t, \boldsymbol{\theta}_{t-1} - \boldsymbol{\theta}^* \rangle - 2\eta_t \beta_{1t} \langle \boldsymbol{m}_{t-1}, \boldsymbol{\theta}_{t-1} - \boldsymbol{\theta}^* \rangle \\
&\leq \|\tilde{\boldsymbol{v}}_t^{1/4}(\boldsymbol{\theta}_{t-1} - \boldsymbol{\theta}^*)\|_2^2 + \eta_t^2 \|\tilde{\boldsymbol{v}}_t^{-1/4} \boldsymbol{m}_t\|_2^2 - 2\eta_t(1 - \beta_{1t})\langle \boldsymbol{g}_t, \boldsymbol{\theta}_{t-1} - \boldsymbol{\theta}^* \rangle \\
&\quad + \eta_t^2 \beta_{1t} \|\tilde{\boldsymbol{v}}_t^{-1/4} \boldsymbol{m}_{t-1}\|_2^2 + \beta_{1t} \|\tilde{\boldsymbol{v}}_t^{1/4}(\boldsymbol{\theta}_{t-1} - \boldsymbol{\theta}^*)\|_2^2,
\end{aligned} \tag{15}
$$

where the above inequality uses the Cauchy-Schwartz inequality $2\langle \boldsymbol{a}, \boldsymbol{b} \rangle \leq \|\boldsymbol{a}\|_2^2 + \|\boldsymbol{b}\|_2^2$. Note that (15) corresponds to (2) in the appendix of Zhuang et al. (2020) for AdaBelief.

Summing (15) from $t = 1$ until $t = T$, rearranging the quantities, and exploiting the property that $\boldsymbol{g}_t = \nabla f(\boldsymbol{\theta}_{t-1})$ and $f(\cdot)$ being convex gives

$$
\begin{aligned}
&f(\bar{\boldsymbol{\theta}}_T) - f(\boldsymbol{\theta}^*) \\
&= f\left(\frac{1}{T}\sum_{t=0}^{T-1}\boldsymbol{\theta}_t\right) - f(\boldsymbol{\theta}^*) \\
&\stackrel{(a)}{\leq} \frac{1}{T}\sum_{t=1}^{T}(f(\boldsymbol{\theta}_{t-1}) - f(\boldsymbol{\theta}^*)) \\
&\stackrel{(b)}{\leq} \frac{1}{T}\sum_{t=1}^{T}\langle \boldsymbol{g}_t, \boldsymbol{\theta}_{t-1} - \boldsymbol{\theta}^* \rangle \\
&\leq \frac{1}{T}\sum_{t=1}^{T}\left[\frac{1}{2\eta_t(1 - \beta_{1t})}\left(\|\tilde{\boldsymbol{v}}_t^{1/4}(\boldsymbol{\theta}_{t-1} - \boldsymbol{\theta}^*)\|_2^2 - \|\tilde{\boldsymbol{v}}_t^{1/4}(\boldsymbol{\theta}_t - \boldsymbol{\theta}^*)\|_2^2\right)\right.
\end{aligned}
$$

$$+ \frac{\eta_t}{2(1-\beta_{1t})}\|\tilde{v}_t^{-1/4}m_t\|_2^2 + \frac{\eta_t\beta_{1t}}{2(1-\beta_{1t})}\|\tilde{v}_t^{-1/4}m_{t-1}\|_2^2 + \frac{\beta_{1t}}{2\eta_t(1-\beta_{1t})}\|\tilde{v}_t^{1/4}(\theta_{t-1}-\theta^*)\|_2^2\Big]$$

$$\overset{\beta_{11}=\beta_1}{=} \frac{1}{2\eta(1-\beta_1)T}\|\tilde{v}_1^{1/4}(\theta_0-\theta^*)\|_2^2$$

$$+ \frac{1}{T}\sum_{t=1}^{T-1}\left(\frac{1}{2\eta_{t+1}(1-\beta_{1(t+1)})}\|\tilde{v}_{t+1}^{1/4}(\theta_t-\theta^*)\|_2^2 - \frac{1}{2\eta_t(1-\beta_{1t})}\|\tilde{v}_t^{1/4}(\theta_t-\theta^*)\|_2^2\right)$$

$$+ \frac{1}{T}\sum_{t=1}^{T}\left[\frac{\eta_t}{2(1-\beta_{1t})}\|\tilde{v}_t^{-1/4}m_t\|_2^2 + \frac{\eta_t\beta_{1t}}{2(1-\beta_{1t})}\|\tilde{v}_t^{-1/4}m_{t-1}\|_2^2 + \frac{\beta_{1t}}{2\eta_t(1-\beta_{1t})}\|\tilde{v}_t^{1/4}(\theta_{t-1}-\theta^*)\|_2^2\right]$$

$$\left(\text{condition: }\left\{\begin{array}{l}0 \le \tilde{v}_t[i] \le \tilde{v}_{t-1}[i] \text{ for all } i=1,\ldots,d, \\ 0 \le \beta_{1(t+1)} \le \beta_{1t} < 1\end{array}\right.\right)$$

$$\le \frac{1}{2\eta(1-\beta_1)T}\|\tilde{v}_1^{1/4}(\theta_0-\theta^*)\|_2^2$$

$$+ \frac{1}{T}\sum_{t=1}^{T-1}\left(\frac{1}{\eta_{t+1}}-\frac{1}{\eta_t}\right)\frac{1}{2(1-\beta_{1t})}\|\tilde{v}_{t+1}^{1/4}(\theta_t-\theta^*)\|_2^2$$

$$+ \frac{1}{T}\sum_{t=1}^{T}\left[\frac{\eta_t}{2(1-\beta_1)}\|\tilde{v}_t^{-1/4}m_t\|_2^2 + \frac{\eta_t\beta_1}{2(1-\beta_1)}\|\tilde{v}_{t-1}^{-1/4}m_{t-1}\|_2^2 + \frac{\beta_{1t}}{2\eta_t(1-\beta_1)}\|\tilde{v}_t^{1/4}(\theta_{t-1}-\theta^*)\|_2^2\right]$$

$$(\text{condition: } \eta_t = \eta/\sqrt{t}, 0 \le \beta_{1t+1} \le \beta_{1t} < 1, m_0 = \mathbf{0})$$

$$\le \frac{1}{2\eta(1-\beta_1)T}\|\tilde{v}_1^{1/4}(\theta_0-\theta^*)\|_2^2 + \frac{1}{2T\eta(1-\beta_1)}\sum_{t=1}^{T-1}\left(\sqrt{t+1}-\sqrt{t}\right)\|\tilde{v}_{t+1}^{1/4}(\theta_t-\theta^*)\|_2^2$$

$$+ \frac{1}{T}\sum_{t=1}^{T}\frac{\eta_t(1+\beta_1)}{2(1-\beta_1)}\|\tilde{v}_t^{-1/4}m_t\|_2^2 + \frac{1}{T(1-\beta_1)}\sum_{t=1}^{T}\frac{\beta_{1t}}{2\eta_t}\|\tilde{v}_t^{1/4}(\theta_{t-1}-\theta^*)\|_2^2$$

$$(\text{condition: }\|\theta^*\|_\infty \le D, \|\theta_t\|_\infty \le D)$$

$$\le \frac{D^2}{\eta(1-\beta_1)T}\sum_{i=1}^{d}(\tilde{v}_1[i])^{1/2} + \frac{D^2}{T\eta(1-\beta_1)}\sum_{t=1}^{T-1}\left(\sqrt{t+1}-\sqrt{t}\right)\sum_{i=1}^{d}(\tilde{v}_{t+1}[i])^{1/2}$$

$$+ \frac{1}{T}\sum_{t=1}^{T}\frac{\eta_t(1+\beta_1)}{2(1-\beta_1)}\|\tilde{v}_t^{-1/4}m_t\|_2^2 + \frac{D^2}{T(1-\beta_1)}\sum_{t=1}^{T}\frac{\beta_{1t}}{\eta_t}\sum_{i=1}^{d}(\tilde{v}_t[i])^{1/2}$$

$$\overset{(c)}{\le} \frac{D^2 d(G_\infty+\sqrt{\epsilon})}{\eta(1-\beta_1)(1-\beta_2)T} + \frac{D^2 d(G_\infty+\sqrt{\epsilon})}{\sqrt{T}\eta(1-\beta_1)(1-\beta_2)} + \frac{1}{T}\sum_{t=1}^{T}\frac{\eta_t(1+\beta_1)}{2(1-\beta_1)}\|\tilde{v}_t^{-1/4}m_t\|_2^2 + \frac{D^2\beta_1 d(G_\infty+\sqrt{\epsilon})}{T(1-\beta_1)(1-\beta_2)\eta(1-\lambda)^2},$$

$$(16)$$

where both step $(a)$ and step $(b)$ use the property of $f(\cdot)$ being convex, and step $(c)$ uses the following conditions

$$\left\{\begin{array}{l}\|g_t\|_\infty \le G_\infty/2 \Rightarrow \|m_{l,t}^{(K)}-g_{l,t}^{(K)}\|_\infty \le G_\infty \Rightarrow \|\tilde{v}_t\|_\infty \le (G_\infty+\sqrt{\epsilon})^2/(1-\beta_2) \\ \sum_{t=1}^{T}\frac{\beta_{1t}}{\eta_t} \le \frac{\beta_1}{\eta}\sum_{t=1}^{T}\lambda^{t-1}\sqrt{t} \le \frac{\beta_1}{\eta}\sum_{t=1}^{T}\lambda^{t-1}t \le \frac{\beta_1}{\eta(1-\lambda)^2},\end{array}\right. \tag{17}$$

where the 1st condition in (17) is derived by using the expression

$$\tilde{v}_{l,t} = \frac{v_{l,t}}{1-\beta_2^t} = \frac{s_{l,t}}{1-\beta_2^t} + \frac{\epsilon}{1-\beta_2} \text{ where } s_{l,t} = (1-\beta_2)s_{l,t-1} + \beta_2(m_{l,t}^{(K)}-g_{l,t}^{(K)})^2.$$

Next we consider the quantity $\sum_{t=1}^{T}\eta_t\|\tilde{v}_t^{-1/4}m_t\|_2^2$ in (16), the upper bound of which is given in Zhuang et al. (2020):

**Lemma 1** (Equ. (4) in the appendix of Zhuang et al. (2020)). *Let $g_{1:T}^2[i] = ((g_1[i])^2,\ldots,(g_T[i])^2) \in \mathbb{R}^T$. Under the three assumptions given in the theorem, we have*

$$\sum_{t=1}^{T}\eta_t\|\tilde{v}_t^{-1/4}m_t\|_2^2 \le \frac{\eta\sqrt{1+\log T}}{\sqrt{c}(1-\beta_1)^2}\|(g_{1:T}^2[i])\|_2. \tag{18}$$

Finally, plugging (18) into (16) produces the upper-bound regret in the theorem. The proof is complete.

## C  Parameter Setups for Optimization Methods in Fig. 1-3

The common stepsize $\eta_t$ is dropped by a factor 0.1 at 100 and 160 epochs. The optimal parameter $\epsilon$ is searched over the set $\{10^{-2}, 10^{-3}, \ldots, 10^{-15}\}$ for Adam and AdaBelief*.

| optimizer | fixed parameters | searched parameters |
|---|---|---|
| Adam | $(\eta_0, \beta_1, \beta_2)$ $= (0.001, 0.9, 0.999)$ | $\epsilon \in \{10^{-2}, 10^{-3}, \ldots, 10^{-15}\}$ |
| AdaBelief* | $(\eta_0, \beta_1, \beta_2)$ $= (0.001, 0.9, 0.999)$ | $\epsilon \in \{10^{-2}, 10^{-3}, \ldots, 10^{-15}\}$ |
| AdaBelief | $(\eta_0, \beta_1, \beta_2, \epsilon)$ $= (0.001, 0.9, 0.999, 10^{-8})$ | |
| Aida ($K=1$) | $(\eta_0, \beta_1, \beta_2, \epsilon)$ $= (0.001, 0.9, 0.999, 10^{-8})$ | |
| Aida ($K=2$) | $(\eta_0, \beta_1, \beta_2, \epsilon)$ $= (0.001, 0.9, 0.999, 10^{-8})$ | |

## D  Parameter-setups for training different DNN models

Table 6:  Parameter-setups for training ResNet18 over ImageNet. The weight decay was set to $10^{-2}$.

| optimizer | fixed parameters | searched parameters |
|---|---|---|
| AdaBelief | $(\eta_0, \beta_1, \beta_2)$ $= (0.001, 0.9, 0.999)$ | $\epsilon = \{10^{-8}, 10^{-9}, 10^{-10}\}$ |
| Aida ($K=2$) | $(\eta_0, \beta_1, \beta_2, \epsilon)$ $= (0.001, 0.9, 0.999, 10^{-9})$ | |

Table 7:  Parameter-setups for training a Transformer. The weight decay for AdamW was set to $5e-4$ while the weight decay for all other algorithms was set to 0.0. The setup $\beta_2 = 0.98$ follows directly from the first open-source repository. One observes that the search grids of $\epsilon$ are different for different optimizers. This is because we stopped searching for a particular optimizer when the performance dropped significantly as $\epsilon$ is either too large or too small.

| optimizer | fixed parameters | searched parameters |
|---|---|---|
| AdaBound | $(\eta_0, \beta_1, \beta_2, \gamma) = (0.001, 0.9, 0.98, 0.001)$ | $\epsilon \in \{1e-6, 1e-7, \ldots, 1e-16\}$ final_stepsize $\in \{0.1, 0.01, 0.001\}$ |
| Yogi | $(\eta_0, \beta_1, \beta_2) = (0.001, 0.9, 0.98)$ | $\epsilon \in \{1e-2, 1e-3, \ldots, 1e-16\}$ |
| SGD | momentum=0.9 | $\eta_0 \in \{1.0, 0.1, 0.01, 0.001\}$ |
| RAdam | $(\eta_0, \beta_1, \beta_2) = (0.001, 0.9, 0.98)$ | $\epsilon \in \{1e-6, 1e-7, \ldots, 1e-16\}$ |
| MSVAG | $(\eta_0, \beta_1, \beta_2) = (0.001, 0.9, 0.98)$ | $\epsilon \in \{1e-6, 1e-7, \ldots, 1e-16\}$ |
| Fromage | | $\eta_0 \in \{0.1, 0.01, 0.001, 0.0001\}$ |
| Adam | $(\eta_0, \beta_1, \beta_2) = (0.001, 0.9, 0.98)$ | $\epsilon \in \{1e-6, 1e-7, \ldots, 1e-16\}$ |
| AdamW | $(\eta_0, \beta_1, \beta_2) = (0.001, 0.9, 0.98)$ | $\epsilon \in \{1e-6, 1e-7, \ldots, 1e-16\}$ |
| AdaBelief | $(\eta_0, \beta_1, \beta_2) = (0.001, 0.9, 0.98)$ | $\epsilon \in \{1e-8, 1e-9, \ldots, 1e-16\}$ |
| Aida(K=1)(**our**) | $(\eta_0, \beta_1, \beta_2, \epsilon) = (0.001, 0.9, 0.98, 1e-16)$ | |
| Aida(K=2)(**our**) | $(\eta_0, \beta_1, \beta_2, \epsilon) = (0.001, 0.9, 0.98, 1e-16)$ | |

Table 8: Parameter-setups for training LSTMs. The weight decay for every algorithm was set to $1.2e - 6$.

| optimizer | fixed parameters | searched parameters |
|---|---|---|
| AdaBound | $(\beta_1, \beta_2, \gamma) = (0.9, 0.999, 0.001)$ | $\eta_0 \in \{0.01, 0.001\}$ 
 $\epsilon \in \{1e - 6, 1e - 7, \dots, 1e - 16\}$ 
 final_stepsize $\in \{0.1, 3, 30\}$ |
| Yogi | $(\beta_1, \beta_2) = (0.9, 0.999)$ | $\eta_0 \in \{0.01, 0.001\}$ 
 $\epsilon \in \{1e - 2, 1e - 3, 1e - 4, 1e - 16\}$ |
| SGD | momentum=0.9 | $\eta_0 \in \{30, 3, 1, 0.1\}$ |
| RAdam | $(\beta_1, \beta_2) = (0.9, 0.999)$ | $\eta_0 \in \{0.01, 0.001\}$ 
 $\epsilon \in \{1e - 6, 1e - 7, \dots, 1e - 16\}$ |
| MSVAG | $(\beta_1, \beta_2) = (0.9, 0.999)$ | $\eta_0 \in \{30, 1, 0.01, 0.001\}$ 
 $\epsilon \in \{1e - 6, 1e - 7, \dots, 1e - 16\}$ |
| Fromage | | $\eta_0 \in \{0.1, 0.01, 0.001\}$ |
| Adam | $(\beta_1, \beta_2) = (0.9, 0.999)$ | $\eta_0 \in \{0.01, 0.001\}$ 
 $\epsilon \in \{1e - 6, 1e - 7, \dots, 1e - 16\}$ |
| AdamW | $(\beta_1, \beta_2) = (0.9, 0.999)$ | $\eta_0 \in \{0.01, 0.001\}$ 
 $\epsilon \in \{1e - 6, 1e - 7, \dots, 1e - 16\}$ |
| AdaBelief | $(\beta_1, \beta_2) = (0.9, 0.999)$ | $\eta_0 \in \{0.01, 0.001\}$ 
 $\epsilon \in \{1e - 8, 1e - 9, \dots, 1e - 16\}$ |
| Aida(K=1) (**our**) | $(\eta_0, \beta_1, \beta_2, \epsilon) = (0.001, 0.9, 0.999, 1e - 16)$ | |
| Aida(K=2) (**our**) | $(\eta_0, \beta_1, \beta_2, \epsilon) = (0.001, 0.9, 0.999, 1e - 16)$ | |

Table 9: Parameter-setups for training WGAN-GP. The weight decay for AdamW was set to $5e - 4$ while the weight decay for all other algorithms was set to 0.0. We stopped searching the parameter $\epsilon$ for a particular adaptive optimizer when the FID score dropped significantly for either too large or too small $\epsilon$ values.

| optimizer | fixed parameters | searched parameters |
|---|---|---|
| AdaBound | $(\eta_0, \beta_1, \beta_2, \gamma) = (0.0002, 0.5, 0.999, 0.001)$ | $\epsilon \in \{1e - 2, 1e - 4, \dots, 1e - 14\}$ 
 final_stepsize $\in \{0.1, 0.01\}$ |
| Yogi | $(\eta_0, \beta_1, \beta_2) = (0.0002, 0.5, 0.999)$ | $\epsilon \in \{1e - 2, 1e - 3, 1e - 4, 1e - 14\}$ |
| SGD | | momentum $= \{0.3, 0.5, 0.9\}$ 
 $\eta_0 \in \{0.1, 0.02, 0.002, 0.0002\}$ |
| RAdam | $(\eta_0, \beta_1, \beta_2) = (0.0002, 0.5, 0.999)$ | $\epsilon \in \{1e - 4, 1e - 6, \dots, 1e - 14\}$ |
| MSVAG | $(\beta_1, \beta_2) = (0.5, 0.999)$ | $\eta_0 \in \{0.1, 0.02, 0.002, 0.0002\}$ 
 $\epsilon \in \{1e - 4, 1e - 6, \dots, 1e - 14\}$ |
| Fromage | | $\eta_0 \in \{0.1, 0.01, 0.001\}$ |
| Adam | $(\eta_0, \beta_1, \beta_2) = (0.0002, 0.5, 0.999)$ | $\epsilon \in \{1e - 4, 1e - 6, \dots, 1e - 14\}$ |
| AdamW | $(\eta_0, \beta_1, \beta_2) = (0.0002, 0.5, 0.999)$ | $\epsilon \in \{1e - 4, 1e - 6, \dots, 1e - 14\}$ |
| AdaBelief | $(\eta_0, \beta_1, \beta_2, \epsilon) = (0.0002, 0.5, 0.999, 1e - 12)$ | |
| Aida(K=1) (**our**) | $(\eta_0, \beta_1, \beta_2, \epsilon) = (0.0002, 0.5, 0.999, 1e - 12)$ | |
| Aida(K=2) (**our**) | $(\eta_0, \beta_1, \beta_2, \epsilon) = (0.0002, 0.5, 0.999, 1e - 12)$ | |

Table 10: Parameter-setups for training VGG and ResNet models over CIFAR10 and CIFAR100. The weight decay for AdamW was set to $0.01$ while the weight decay for all other algorithms was set to $5e-4$. As we mentioned in the main paper, we stopped searching the parameter $\epsilon$ when the validation performance dropped significantly for either too large or too small $\epsilon$ values.

| optimizer | fixed-parameters | searched-parameters |
|---|---|---|
| AdaBound | $(\eta_0, \beta_1, \beta_2, \gamma) = (0.001, 0.9, 0.999, 0.001)$ | $\epsilon \in \{1e-2, 1e-3, \ldots, 1e-9\}$ 
 final_stepsize $\in \{0.1, 0.01\}$ |
| Yogi | $(\eta_0, \beta_1, \beta_2,) = (0.001, 0.9, 0.999)$ | $\epsilon \in \{1e-1, 1e-2, \ldots, 1e-9\}$ |
| SGD | $(\text{momentum}, \eta_0) = (0.9, 0.1)$ | |
| RAdam | $(\eta_0, \beta_1, \beta_2) = (0.001, 0.9, 0.999)$ | $\epsilon \in \{1e-2, 1e-3, \ldots, 1e-9\}$ |
| MSVAG | $(\eta_0, \beta_1, \beta_2) = (0.1, 0.9, 0.999)$ | $\epsilon \in \{1e-2, 1e-2, \ldots, 1e-9\}$ |
| Fromage | | $\eta_0 \in \{0.1, 0.01, 0.001\}$ |
| Adam | $(\eta_0, \beta_1, \beta_2) = (0.001, 0.9, 0.999)$ | $\epsilon \in \{1e-2, 1e-3, \ldots, 1e-9\}$ |
| AdamW | $(\eta_0, \beta_1, \beta_2) = (0.001, 0.9, 0.999)$ | $\epsilon \in \{1e-2, 1e-3, \ldots, 1e-9\}$ |
| AdaBelief | $(\eta_0, \beta_1, \beta_2) = (0.001, 0.9, 0.999)$ | $\epsilon \in \{1e-8, 1e-9, 1e-10\}$ |
| Aida(K=1) (**our**) | $(\eta_0, \beta_1, \beta_2, \epsilon) = (0.001, 0.9, 0.999, 1e-9)$ | |
| Aida(K=2) (**our**) | $(\eta_0, \beta_1, \beta_2, \epsilon) = (0.001, 0.9, 0.999, 1e-9)$ | |

---

**Algorithm 3** Aida$^*$: Scaling $(\mathbf{m}_{l,t}, \mathbf{g}_{l,t})$ via power of $\cos \angle \mathbf{m}_{l,t} \mathbf{g}_{l,t}$

---

**Input:** $\beta_1, \beta_2, \eta_t, \epsilon > 0, \xi = 1e-20, K = 2$
**Init.:** $\boldsymbol{\theta}_0 \in \mathbb{R}^d$, $\boldsymbol{m}_0 = 0$, $\boldsymbol{v}_0 = \tilde{\boldsymbol{v}}_0 = 0 \in \mathbb{R}^d$
**for** $t = 1, 2, \ldots, T$ **do**
  $\boldsymbol{g}_t \leftarrow \nabla f(\boldsymbol{\theta}_{t-1})$
  $\boldsymbol{m}_t \leftarrow \beta_1 \boldsymbol{m}_{t-1} + (1 - \beta_1) \boldsymbol{g}_t$
  **for** $l = 1, \ldots, L$ **do**
    $\gamma_{l,t} = \langle \boldsymbol{m}_{l,t}, \boldsymbol{g}_{l,t} \rangle / (\|\boldsymbol{m}_{l,t}\| \|\boldsymbol{g}_{l,t}\| + \xi)$
    $\boldsymbol{v}_{l,t} \leftarrow \beta_2 \boldsymbol{v}_{l,t-1} + (1 - \beta_2)(\gamma_{l,t}^K \boldsymbol{m}_{l,t} - \gamma_{l,t}^K \boldsymbol{g}_{l,t})^2 + \epsilon$    [the $K$th power of $\gamma_{l,t}$ is employed for scaling]
  **end for**
  $\tilde{\boldsymbol{m}}_t \leftarrow \frac{\boldsymbol{m}_t}{1-\beta_1^t}$   $\left\{ \tilde{\boldsymbol{v}}_{l,t} \leftarrow \frac{\boldsymbol{v}_{l,t}}{1-\beta_2^t} \right\}_{l=1}^L$
  $\boldsymbol{\theta}_t \leftarrow \boldsymbol{\theta}_{t-1} - \frac{\eta_t \tilde{\boldsymbol{m}}_t}{\sqrt{\tilde{\boldsymbol{v}}_t}}$
**end for**
**Output:** $\boldsymbol{\theta}_T$

---

# E   Performance Investigation by Scaling $(\mathrm{m}_{l,t}, \mathrm{g}_{l,t})$ via power of $\cos \angle \mathrm{m}_{l,t} \mathrm{g}_{l,t}$

In this section, we consider scaling $\boldsymbol{g}_{l,t}$ and $\boldsymbol{m}_{l,t}$ in the computation of the second momentum in AdaBelief by utilizing the power of $\cos \angle \boldsymbol{g}_{l,t} \boldsymbol{m}_{l,t}$, which is computed as

$$\cos \angle \boldsymbol{g}_{l,t} \boldsymbol{m}_{l,t} = \gamma_{l,t} = \langle \boldsymbol{g}_{l,t}, \boldsymbol{m}_{l,t} \rangle / (\|\boldsymbol{g}_{l,t}\| \|\boldsymbol{m}_{l,t}\| + \xi), \tag{19}$$

where the parameter $\xi$ is introduced to avoid division by zero. The modified optimizer is referred to as Aida$^*$ (see Alg. 3 above for the complete update procedure). The only difference between Aida and Aida$^*$ is the scaling of the layerwise vectors $(\boldsymbol{m}_{l,t}, \boldsymbol{g}_{l,t})$.

We performed additional experiments by applying Aida$^*$ to train ResNet34 and LSTMs. The performance results are provided in Table 11 and 12, respectively. It is seen that the validation accuracies (the higher the better) for training ResNet34 over both CIFAR10 and CIFAR100 are degraded slightly in comparison to Aida. Also, the test perplexities (the lower the better) for training 1-layer and 3-layer LSTMs are slightly degraded. The above experiments suggest that the parameter $\xi$ in (10) and (9) in the mutual-vector projections does indeed make a difference in performance. In our work, we recommend the implementation in Alg. 1.

Table 11: Performance comparison of Aida and Aida* for training ResNet34. The values in the table represent validation accuracies (the higher the better).

| optimizers | CIFAR10 | CIFAR100 |
|---|---|---|
| Aida ($K = 2$) | **95.57**±0.13 | **78.86**±0.12 |
| Aida* ($K = 2$) | 95.43±0.12 | 78.44±0.34 |

Table 12: Performance comparison of Aida and Aida* for training LSTMs. The values in the table represent test perplexity (the lower the better).

| optimizers | 1-layer | 2-layer | 3-layer |
|---|---|---|---|
| Aida ($K = 2$) | **81.53** | 65.04 | **60.18** |
| Aida* ($K = 2$) | 81.58 | **64.85** | 60.43 |

# F   Additional Plots of Learning Curves Versus Runtime

The two figures Fig. 5 and 6 only plot the learning curves of Adam, AdaBelief, and Aida over epochs. Since Aida needs additional computational overhead compared to AdaBelief, it is interesting to also study the learning curves of Adam, AdaBelief, and Aida over runtime. Relevant results are shown in Fig. 7 and 8. It is clear that even though Aida requires a slightly longer learning time, the method produces significant validation performance gain for training both transformers and LSTMs.

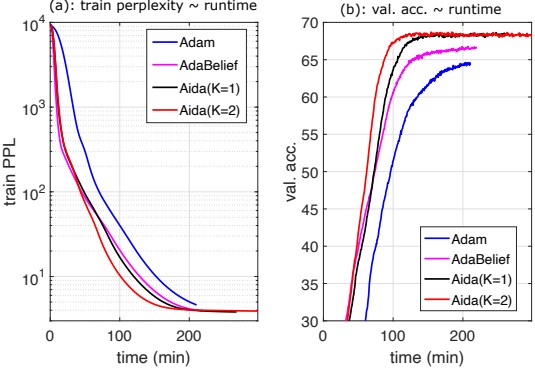

Figure 7: Visualisation of the learning curves versus runtime for Aida, AdaBelief, and Adam when training the Transformer. The maximum number of epochs is set to 400 in all three methods. The execution time for each epoch takes into account both the training and validation time.

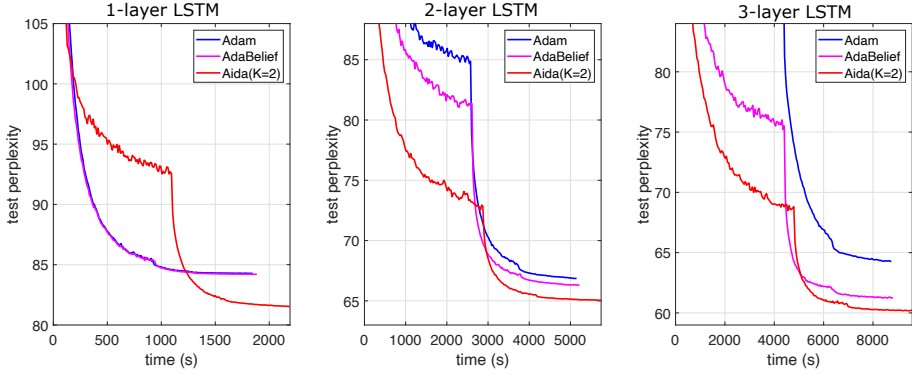

Figure 8: Visualisation of the learning curves versus runtime for Aida, AdaBelief, and Adam when training LSTMs. The maximum number of epochs is set to 200 in all three methods. The execution time for each epoch takes into account both the training and validation time.

