# OpenReview forum: "A DNN Optimizer that Improves over AdaBelief by Suppression of the Adaptive Stepsize Range"
_TMLR — Accepted by TMLR_

### Review · Reviewer_17x9 · 2023-04-25

**Summary Of Contributions:**

This paper aims at improving adaptive step size optimizers. Inspired by previous work that links compact adaptive step size range to better generalization, the paper proposes a modification to the AdaBelief that suppressed the range of its adaptive step size. It is achieved by adding a scaling term to both the momentum and gradient term when updating the state variable for the adaptive factor (denominator); the scaling term is computed using mutual projection between two vectors, which gradually reduces the length of each vector while preserving their angle. A convergence analysis is provided to support this change. Empirical evaluation is conducted on a range of tasks and models: image classification (VGG ResNet), generation (WGAN), and machine translation (Transformer, LSTM), where the proposed optimizer Aida achieves consistent improvement over baseline optimizers, with 25% computational overhead (on some tasks).

**Audience:**

Yes

**Claims And Evidence:**

Yes

**Requested Changes:**

1. Include a discussion on the criteria of hyperparameter choice for each experiment.
2. Also plot the learning curve against runtime to account for the 25% computation overhead.

**Strengths And Weaknesses:**

Strength:
1. The proposed modification to AdaBelief is well motivated, directly following the observation from previous work, that a compact adaptive step size range helps with improving the generalization.
2. Empirical results across different tasks, models, and datasets show that the resulting method achieves consistent gain over existing optimizers, and also speedup the convergence. Yet I have some doubts regarding the hyperparameter choice of those experiments.

Weakness:
1. Lack of justification for the choice of hyperparameters for each optimizer in the experiments. While the hyper-parameters for each optimizer are provided in the appendix, it is not always clear to me why they are set that way, how they are selected, and whether their choice proposes fair comparison. For example, the epsilon of Aida for each task is different.; sometimes the epsilon for baselines are searched while the learning rates are fixed; The first momentum in WGAN training is different than others; The choice of which hyperparameters to fix and which hyperparameters to search misses explanation; The search grid of hyperparameters are different across different tasks. I would appreciate more elaboration on that.
2. Non-negligible computational overhead. Since the computational overhead of Aida over Adabelief is 25%, it might be worthwhile to also plot the learning curve against runtime to account for that.

---

> ### Author Response · Authors · 2023-05-08
>
>
> 1. "Lack of justification for the choice of hyperparameters for each optimizer in the experiments. While the hyper-parameters for each optimizer are provided in the appendix, it is not always clear to me why they are set that way, how they are selected, and whether their choice proposes fair comparison. For example, the epsilon of Aida for each task is different.; sometimes the epsilon for baselines are searched while the learning rates are fixed; The first momentum in WGAN training is different than others; The choice of which hyperparameters to fix and which hyperparameters to search misses explanation; The search grid of hyperparameters are different across different tasks. I would appreciate more elaboration on that."
>
> Thank you for raising concerns regarding the selection of the hyper-parameters. Since our new algorithm Aida is an extension of the AdaBelief, we exploited the open-source repository of AdaBelief to evaluate the performance of Aida. The tested hyper-parameters are inherited from the open-source code for AdaBelief, which was used to compare AdaBelief with other eight optimizers (including SGD, RAdam, AdaBound, Yogi, AdamW, MSVAG, Fromage, and Adam).
>
> To be specific, in the open-source of AdaBelief, the parameter $\epsilon$ of AdaBelief was set differently for different DNN tasks. The setting for the parameter $\epsilon$ in Aida follows that of the Adabelief setup and hence also was configured differently for different tasks. The setting $\beta_1$=0.5 in WGAN training also follows the corresponding setting of AdaBelief and is a typical setting in general GAN-training.  In the open-source of AdaBelief, the initial learning rate of the reference optimizers was searched together with $\epsilon$ for LSTM. To make a fair comparison in our work, we also searched the initial learning rate for those optimizers when training LSTM. In general, given an optimizer, the optimal parameter $\epsilon$ is different for different DNN tasks. In our experiments, we stopped searching when we observed that the performance decreased as $\epsilon$ is either too large or too small. This is why the search grid is different for different tasks.
>
> 2. "Non-negligible computational overhead. Since the computational overhead of Aida over Adabelief is 25 \%, it might be worthwhile to also plot the learning curve against runtime to account for that."
>
> Thank you for the good suggestions. In addition to figures 5 and 6 in the paper, we will also plot their learning curves against run times in the revision. We plan to put the new figures into the appendix, and briefly mention the new figures when we discuss figures 5 and 6 in the main paper.

---

### Review · Reviewer_cU6d · 2023-04-27

**Summary Of Contributions:**

The paper investigates the AdaBelief algorithm, specifically how a small constant added to the squared gradient moving average equation (middle eq in Eq 3) in AdaBelief significantly impacts optimization and generalization. Based on the gathered insight, the authors then propose an additional change to the AdaBelief algorithm, and call the new algorithm Aida. A convergence proof is shown for the proposed algorithm. Finally, the authors show experiments using Aida and previous optimizers and claim improved optimization and generalization performance.

**Audience:**

Yes

**Claims And Evidence:**

Yes

**Requested Changes:**

see above

It would also be helpful to have a discussion on why reducing the adaptive step size range improves optimization and generalization.

**Strengths And Weaknesses:**

Strengths:
- The proposed optimizer achieves better generalization and faster convergence in some cases.
- The analysis of AdaBelief is interesting.

Weakness:
- The motivation behind the specific approach of iterative projection, proposed by the authors for their method Aida, is unclear. Specifically, after making the observation that the epsilon term in AdaBelief helps keep the adaptive step size range small (which helps generalization), the authors propose to perform this iterative projection of the g and m (gradient and the moving average of gradient respectively) onto each other. The resulting g and m are in the same direction as the original g or m (depending on the number of iterative projection steps). Importantly, the result of this iterative projection is that the scale of m and g becomes smaller in magnitude, which the authors also discuss in section 3.1. In fact, the iterative projections are computationally redundant, since the cosine of the angle between m and g during this iterative process remains unchanged. Therefore, I wonder why the authors chose to propose this iterative projection approach, as opposed to (say) scaling g and m using a scalar hyper-parameter multiplied with the cosine of the angle between g and m. That would've been both simpler and computationally cheaper.

- My above point is further strengthened by the fact that the authors find that K=2 already yields good results (where K is the number of iterative projection steps). I understand that once advantage of using this iterative approach instead of directly specifying a scaling hyper-parameter is that K=2 is a much simpler choice over choosing a floating number as a hyper-parameter. If the authors choose to use this justification, then I suggest the following changes:
 1. replace the dot product in the iterative projection step with the equivalent vector norms times cosine, which does not require redundant computation.
 2. Explicitly mention the reasoning for proposing this approach as described above.
 Note that while a convergence analysis is provided for the proposed approach, it merely shows sufficiency, and does not imply that the proposed iterative approach is a necessary one.

- In almost all plots, Aida converges faster than other methods. However, in figure 6a Aida has a much slower convergence initially. This was a bit concerning to me.

---

> ### Author Response · Authors · 2023-05-08
>
>
> 1. "The motivation behind the specific approach of iterative projection, proposed by the authors for their method Aida, is unclear. Why not scaling g and m using power of the cosine of the angle between g and m?"
>
> Many thanks for the insightful comments. In principle, the two vectors $g_l$ and $m_l$ for the $l$'th neural layer can be scaled by the power of the cosine of the angle between $g_l$ and $m_l$, which is equivalent to two ($K=2$) mutual-vector projections between $g_l$ and $m_l$ in our paper if the parameter $\xi$ is ignored in (9) and (10). In practice, we found that a positive value of the parameter $\xi$ in (9) and (10) in the mutual-vector projections improves the generalization performance, which, for simplicity, was set to $1e-20$ across all the experiments in the paper.
>
> To support our above statement,  we have done additional experiments where we scaled $g_l$ and $m_l$ with the power of cosine of the angle between $g_l$ and $m_l$ by ignoring the parameter $\xi$ in (9) and (10). It was found that the validation accuracy (the higher the better) for training ResNet34 over CIFAR100 was degraded from 78.86 to 78.44. The validation accuracy for training ResNet34 over CIFAR10 was degraded from 95.57 to 95.43.  The test perplexity (the lower the better) for training 1-layer and 3-layer LSTMs was also slightly degraded (e.g., for the 3-layer LSTM, the test perplexity was degraded from 60.18 to 60.43). We emphasize that the degraded performance is still better than that of the eight adaptive optimizers. The above experiments suggest that the parameter $\xi$ in (9) and (10) in the mutual-vector projections does indeed make a difference in performance.
>
> Since the empirical study in our paper shows that Aida with $K=2$ only introduces a small computational cost compared to AdaBelief, we think the formulation of mutual-vector projections with the embedded $\xi$ may be the better choice.
>
> 2. "In almost all plots, Aida converges faster than other methods. However, in figure 6a Aida has a much slower convergence initially. This was a bit concerning to me."
>
> Thank you for your comments. There is an explanation for the learning curves of Aida in Figure 6.  For the experiment of LSTM, the hyper-parameters $(\eta_0, \epsilon)$ of Aida was set to $(\eta_0, \epsilon)=(0.001,1e-16)$ for 1-layer, 2-layer, and 3-layer LSTMs. On the other hand, the optimal hyper-parameters $(\eta_0, \epsilon)$ of AdaBelief are different for different numbers of neural layers. In particular, the optimal setups in AdaBelief are:  $(\eta_0, \epsilon)=(0.001,1e-16)$ for 1-layer LSTM, $(\eta_0, \epsilon)=(0.01,1e-12)$ for 2-layer LSTM, and $(\eta_0, \epsilon)=(0.01,1e-12)$ for 3-layer LSTM. When the hyper-parameters of AdaBelief for the 2-layer and 3-layer LSTMs are set to $(\eta_0, \epsilon)=(0.001,1e-16)$ to be in line with the setup of Aida, we observe consistently that Aida initially converges slower than AdaBelief. We will make an explanation in the revision.
>
> 3. " regarding the discussion on why reducing the adaptive step size range improves optimization and generalization."
>
> Thank you for the comments. Please see our response to the question of "why it is desirable to have a smaller statistical variance of adaptive stepsizes within each neural layer. ..." posted by Reviewer CKQE.

---

### Review · Reviewer_CKQE · 2023-04-30

**Summary Of Contributions:**

The paper proposes a novel optimizer to reduce the range of adaptive stepsizes in each layer using a layerwise vector projection-based method. The authors evaluate the proposed method on various tasks, such as image classification, image generalization, and NLP, and demonstrate that it outperforms some popular optimizers.

**Audience:**

Yes

**Claims And Evidence:**

Yes

**Requested Changes:**

Typo:
The definition of g after Eq 2 : g = \nabla f(\theta)


**Strengths And Weaknesses:**

Strength:

The proposed method is very easy to follow and I think the main difference is from the projection of g and m.
The idea about vector projections between the gradient g_t and first momentum m_t is also very interesting. If related work exists, the authors could provide some discussion in the revision.
The paper also evaluates the proposed in some different tasks, such as computer vision (image classification, image generalization) and NLP.

Weakness:

The motivation is not very clear for me. For example, why we need to suppress the range of adaptive stepsizes. As you mentioned, Adbelief tried to solve this problem, and why you also try to solve it. What is the main problem in Adabelief you mainly focus on and try to solve.
Some claim need more explanation. For example, “The new method has the nice property that the adaptive stepsizes within each neural layer have smaller statistical variance” My question is why we hope the adaptive stepsizes in each layer has a smaller variance. As the adaptive stepsize for different parameters in the same layer are naturally different.
Some related layer-wise optimizers have been proposed, and the authors could discuss these methods in the revision.
The improvement of Aida on ImageNet is not significant, and it's unclear whether the proposed method can work well in large-scale datasets. The authors could provide some discussion on this. Additionally, the reported accuracy of AdaBelief in this paper is lower than the result in the original paper.

---

> ### Author Response · Authors · 2023-05-08
>
> 1."The motivation is not very clear for me. For example, why we need to suppress the range of adaptive stepsizes. As you mentioned, Adabelief tried to solve this problem, and why you also try to solve it. What is the main problem in Adabelief you mainly focus on and try to solve."
>
> Thank you for your comments. The AdaBelief paper introduces the parameter $\epsilon$ when computing the second momentum $s_t$ in (3), without providing any motivation. The main focus of the AdaBelief paper is to motivate the exponential moving average (EMA) of $(s_t-g_t)^2$ instead of the EMA of  $g_t^2$ that is employed in Adam. To our best knowledge, we are the first to show that including the epsilon parameter in the computation of the 2nd momentum in AdaBelief actually suppresses the range of adaptive stepsizes; this is missing in the AdaBelief paper.
>
> Based on our above findings, we then extended AdaBelief by suppressing the range of adaptive stepsizes of AdaBelief to investigate if the generalisation performance can be further improved. The new optimizer Aida is then designed by following the above motivation.
>
>
> 2. "Regarding the explanation why it is desirable to have a smaller statistical variance of adaptive stepsizes within each neural layer. Some related layer-wise optimizers have been proposed, and the authors could discuss these methods in the revision. "
>
> Many thanks for your comments. Intuitively speaking, when the variance of the adaptive stepsizes within a particular neural layer is encouraged to be small by layer-wise manipulation, the update for the model parameters within the same neural layer become relatively robust to elementwise gradient outliers (exploding or vanishing gradient elements across iterations).
>
> The recent paper [a] below proposed two optimizers LARS and LAMB, which are extensions of SGD with momentum and Adam, respectively, by introducing layerwise normalisation in the update of a DNN model per iteration. For LARS, the layerwise momentum $m_l^t$ at the $l$th layer is first normalized by the $l2$ norm and is then used to update the model. According to the paper, it is intuitive that such a
> normalization provides robustness to exploding layerwise gradients and plateaus when the batchsize becomes sufficiently large to be able to alleviate bias effect.
>
> From a high-level point of view, the three algorithms Aida, LARS, and LAMB introduce certain layerwise operations to bring robustness to existing optimizers. In general, the layerwise operations in LAMB (which is an extension of Adam) are more restrictive than those of Aida: in LAMB both the first and second momentum are involved in layerwise normalisation.
>
> [a] Yang You, et al,
> "Large Batch Optimisation for Deep Learning: Training Bert in 76 Minutes", ICLR, 2020.
>
> 3. "Regarding the limited improvement of Aida on ImageNet"
>
> We agree that for the ImageNet experiment, Aida produces a limited improvement compared to AdaBelief. On the other hand, from an overall perspective, all five experiments in our paper demonstrate that Aida consistently outperforms the other eight adaptive optimizers, where the performance gain is significant for certain tasks and small for other tasks. This suggests that proper manipulation of the range of adaptive stepsizes of an adaptive optimizer makes a positive impact on the generalization performance. We hope our work will shed light on the design of new techniques to improve the performance of other existing optimizers by manipulating the range of their adaptive stepsizes in the future.
>
> 4. "Regarding the reported accuracy of AdaBelief for ImageNet is lower than the result in the original paper."
>
> We clarify that we exploited the open-source of AdaBelief for all the experiments except for the training of Transformers. As we mentioned in the paper, the parameter $\epsilon$ for AdaBelief for ImageNet was searched from the set $\{10^{-8},10^{-9},10^{-10} \}$ to give the best validation performance while the open-source of AdaBelief recommends $\epsilon=10^{-8}$
>  (see \url{https://github.com/juntang-zhuang/Adabelief-Optimizer/blob/update_0.2.0/PyTorch_Experiments/imagenet/run.sh}). Furthermore, we repeated the experiment three times to account for random DNN initialisation. In contrast, in the AdaBelief paper, we could not find any evidence that the experiments were repeated for ImageNet. We suspect the lower performance might be due to different versions of the Python packages, which are difficult to track. We note that for the WGAN-GP experiment, better performance was obtained for AdaBelief.

---

### Decision · Action_Editors · 2023-06-09

**Recommendation:** Accept with minor revision

**Comment:**

The paper was appreciated by the reviewers that still raised several questions. The authors replied to those questions well and the reviewer are now leaning toward an acceptance. But the authors discussed several interesting things with the reviewers that were not reported in the revision of the document so the final decision is "accept with minor revision" so that the authors can integrate those discussions (and references) in the final version of the paper (and supp).

For the next revision the authors must take into account all the comments from the reviewers and integrate to the paper the very interesting discussions raised by the reviewers. More details below:
+ (CKQE.1) the motivation detailed in the reply is very clear and should appear in the introduction.
+ (CKQE.2 and cU6d.3) This discussion about the variance is also important and should be in the paper, so should be the new reference used by teh authors in the discussion.
+ (CKQE.3 and 17x9.1) The authors must discuss more in details in the paper the choices of hyper-parameters and why they might not have the same performances as in the original papers.
+ (cU6d.2) discussion about K=2 and cosine between the vector should be in the text and the new results should be provide in the supplementary.
+ (17x9.2) add the learning VS runtime curve as stated in the reply.




**Audience:**

The paper is of interest to the TMLR audience since it provides a better understanding on existing stochastic optimization algorithms and a new one that can be of interest in many applications.

**Claims And Evidence:**

The paper propose a new alternative to AdaBelief that suppresses the range of the adaptive stepsizes. They provide  an interesting discussion  of the impact of $\varepsilon$ on AdaBelief, propose their alternative and a convergence analysis of this algorithm. Numerical experiments are done a a variety of machine learning optimization tasks and show the interest of the method hence validating the claims.